# Neural Stored-program Memory

**Hung Le, Truyen Tran and Svetha Venkatesh**
Applied AI Institute, Deakin University, Geelong, Australia
{lethai,truyen.tran,svetha.venkatesh}@deakin.edu.au

## Abstract

Neural networks powered with external memory simulate computer behaviors. These models, which use the memory to store data for a neural controller, can learn algorithms and other complex tasks. In this paper, we introduce a new memory to store *weights* for the controller, analogous to the stored-program memory in modern computer architectures. The proposed model, dubbed Neural Stored-program Memory, augments current memory-augmented neural networks, creating differentiable machines that can switch programs through time, adapt to variable contexts and thus resemble the Universal Turing Machine. A wide range of experiments demonstrate that the resulting machines not only excel in classical algorithmic problems, but also have potential for compositional, continual, few-shot learning and question-answering tasks.

## 1 Introduction

Recurrent Neural Networks (RNNs) are Turing-complete (Siegelmann & Sontag, 1995). However, in practice RNNs struggle to learn simple procedures as they lack explicit memory (Graves et al., 2014; Mozer & Das, 1993). These findings have sparked a new research direction called Memory Augmented Neural Networks (MANNs) that emulate modern computer behavior by detaching memorization from computation via memory and controller network, respectively. MANNs have demonstrated significant improvements over memory-less RNNs in various sequential learning tasks (Graves et al., 2016; Le et al., 2018a; Sukhbaatar et al., 2015). Nonetheless, MANNs have barely simulated general-purpose computers.

Current MANNs miss a key concept in computer design: stored-program memory. The concept has emerged from the idea of Universal Turing Machine (UTM) (Turing, 1936) and further developed in Harvard Architecture (Broesch, 2009), Von Neumann Architecture (von Neumann, 1993). In UTM, both data and programs that manipulate the data are stored in memory. A control unit then reads the programs from the memory and executes them with the data. This mechanism allows flexibility to perform universal computations. Unfortunately, current MANNs such as Neural Turing Machine (NTM) (Graves et al., 2014), Differentiable Neural Computer (DNC) (Graves et al., 2016) and Least Recently Used Access (LRUA) (Santoro et al., 2016) only support memory for data and embed a single program into the controller network, which goes against the stored-program memory principle.

Our goal is to advance a step further towards UTM by coupling a MANN with an external program memory. The program memory co-exists with the data memory in the MANN, providing more flexibility, reuseability and modularity in learning complicated tasks. The program memory stores the weights of the MANN's controller network, which are retrieved quickly via a key-value attention mechanism across timesteps yet updated slowly via backpropagation. By introducing a meta network to moderate the operations of the program memory, our model, henceforth referred to as Neural Stored-program Memory (NSM), can learn to switch the programs/weights in the controller network appropriately, adapting to different functionalities aligning with different parts of a sequential task, or different tasks in continual and few-shot learning.

To validate our proposal, the NTM armed with NSM, namely Neural Universal Turing Machine (NUTM), is tested on a variety of synthetic tasks including algorithmic tasks from

Graves et al. (2014), composition of algorithmic tasks and continual procedure learning. For these algorithmic problems, we demonstrate clear improvements of NUTM over NTM. Further, we investigate NUTM in few-shot learning by using LRUA as the MANN and achieve notably better results. Finally, we expand NUTM application to linguistic problems by equipping NUTM with DNC core and achieve competitive performances against state-of-the-arts in the bAbI task (Weston et al., 2015).

Taken together, our study advances neural network simulation of Turing Machines to neural architecture for Universal Turing Machines. This develops a new class of MANNs that can store and query both the weights and data of their own controllers, thereby following the stored-program principle. A set of five diverse experiments demonstrate the computational universality of the approach.

## 2 Background

In this section, we briefly review MANN and its relations to Turing Machines. A MANN consists of a controller network and an external memory $\mathbf{M} \in \mathbb{R}^{N \times M}$, which is a collection of $N$ $M$-dimensional vectors. The controller network is responsible for accessing the memory, updating its state and optionally producing output at each timestep. The first two functions are executed by an interface network and a state network[1], respectively. Usually, the interface network is a Feedforward neural network whose input is $c_t$ - the output of the state network implemented as RNNs. Let $W^c$ denote the weight of the interface network, then the state update and memory control are as follows,

$$h_t, c_t = RNN\left(\left[x_t, r_{t-1}\right], h_{t-1}\right) \qquad (1) \qquad\qquad \xi_t = c_t W^c \qquad (2)$$

where $x_t$ and $r_{t-1}$ are data from current input and the previous memory read, respectively. The interface vector $\xi_t$ then is used to read from and write to the memory $\mathbf{M}$. We use a generic notation $memory\left(\xi_t, \mathbf{M}\right)$ to represent these memory operations that either update or retrieve read value $r_t$ from the memory. To support multiple memory accesses per step, the interface network may produce multiple interfaces, also known as control heads. Readers are referred to App. F and Graves et al. (2014; 2016); Santoro et al. (2016) for details of memory read/write examples.

A deterministic one-tape Turing Machine can be defined by 4-tuple $(Q, \Gamma, \delta, q_0)$, in which $Q$ is finite set of states, $q_0 \in Q$ is an initial state, $\Gamma$ is finite set of symbol stored in the tape (the data) and $\delta$ is the transition function (the program), $\delta : Q \times \Gamma \rightarrow \Gamma \times \{-1, 1\} \times Q$. At each step, the machine performs the transition function, which takes the current state and the read value from the tape as inputs and outputs actions including writing new values, moving tape head to new location (left/right) and jumping to another state. Roughly mapping to current MANNs, $Q$, $\Gamma$ and $\delta$ map to the set of the controller states, the read values and the controller network, respectively. Further, the function $\delta$ can be factorized into two sub functions: $Q \times \Gamma \rightarrow \Gamma \times \{-1, 1\}$ and $Q \times \Gamma \rightarrow Q$, which correspond to the interface and state networks, respectively.

By encoding a Turing Machine into the tape, one can build a UTM that simulates the encoded machine (Turing, 1936). The transition function $\delta_u$ of the UTM queries the encoded Turing Machine that solves the considering task. Amongst 4 tuples, $\delta$ is the most important and hence uses most of the encoding bits. In other words, if we assume that the space of $Q$, $\Gamma$ and $q_0$ are shared amongst Turing Machines, we can simulate any Turing Machine by encoding only its transition function $\delta$. Translating to neural language, if we can store the controller network into a queriable memory and make use of it, we can build a Neural Universal Turing Machine. Using NSM is a simple way to achieve this goal, which we introduce in the subsequent section.

---

[1]Some MANNs (e.g., NTM with Feedforward Controller) neglect the state network, only implementing the interface network and thus analogous to one-state Turing Machine.

## 3 Methods

### 3.1 Neural Stored-program Memory

A Neural Stored-program Memory (NSM) is a key-value memory $\mathbf{M}_p \in \mathbb{R}^{P \times (K+S)}$, whose values are the basis weights of another neural network—the programs. $P$, $K$, and $S$ are the number of programs, the key space dimension and the program size, respectively. This concept is a hybrid between the traditional slow-weight and fast-weight (Hinton & Plaut, 1987). Like slow-weight, the keys and values in NSM are updated gradually by backpropagation. However, the values are dynamically interpolated to produce the working weight on-the-fly during the processing of a sequence, which resembles fast-weight computation. Let us denote $\mathbf{M}_p(i).k$ and $\mathbf{M}_p(i).v$ as the key and the program of the $i$-th memory slot. At timestep $t$, given a query key $k_t^p$, the working program is retrieved as follows,

$$D\left(k_t^p, \mathbf{M}_p(i).k\right) = \frac{k_t^p \cdot \mathbf{M}_p(i).k}{||k_t^p|| \cdot ||\mathbf{M}_p(i).k)||} \tag{3}$$

$$w_t^p(i) = \text{softmax}\left(\beta_t^p D\left(k_t^p, \mathbf{M}_p(i).k\right)\right) \tag{4}$$

$$p_t = \sum_{i=1}^{P} w_t^p(i)\,\mathbf{M}_p(i).v \tag{5}$$

where $D(\cdot)$ is cosine similarity and $\beta_t^p$ is the scalar program strength parameter. The vector working program $p_t$ is then reshaped to its matrix form and ready to be used as the weight of other neural networks.

The key-value design is essential for convenient memory access as the size of the program stored in $\mathbf{M}_p$ can be millions of dimensions and thus, direct content-based addressing as in Graves et al. (2014; 2016); Santoro et al. (2016) is infeasible. More importantly, we can inject external control on the behavior of the memory by imposing constraints on the key space. For example, program collapse will happen when the keys stored in the memory stay close to each other. When this happens, $p_t$ is a balanced mixture of all programs regardless of the query key and thus having multiple programs is useless. We can avoid this phenomenon by minimizing a regularization loss defined as the following,

$$l_p = \sum_{i=1}^{P}\sum_{j=i+1}^{P} D\left(\mathbf{M}_p(i).k, \mathbf{M}_p(j).k\right) \tag{6}$$

### 3.2 Neural Universal Turing Machine

It turns out that the combination of MANN and NSM approximates a Universal Turing Machine (Sec. 2). At each timestep, the controller in MANN reads its state and memory to generate control signal to the memory via the interface network $W^c$, then updates its state using the state network $RNN$. Since the parameters of $RNN$ and $W^c$ represent the encoding of $\delta$, we should store both into NSM to completely encode an MANN. For simplicity, in this paper, we only use NSM to store $W^c$, which is equivalent to the Universal Turing Machine that can simulate any one-state Turing Machine.

In traditional MANN, $W^c$ is constant across timesteps and only updated slowly during training, typically through backpropagation. In our design, we compute $W_t^c$ from NSM for every timestep and thus, we need a program interface network—the meta network $P_{\mathcal{I}}$—that generates an interface vector for the program memory: $\xi_t^p = P_{\mathcal{I}}(c_t)$, where $\xi_t^p = [k_t^p, \beta_t^p]$. Together with the $RNN$, $P_{\mathcal{I}}$ simulates $\delta_u$ of the UTM and is implemented as a Feedforward neural network. The procedure for computing $W_t^c$ is executed by following Eqs. (3)-(5), hereafter referred to as $NSM\left(\xi_t^p, \mathbf{M}_p\right)$. Figure 1 depicts the integration of NSM into MANN.

In this implementation, key-value NSM offers a more flexible learning scheme than direct attention, in which the meta-network can generate the weight $w_t^p$ directly without matching

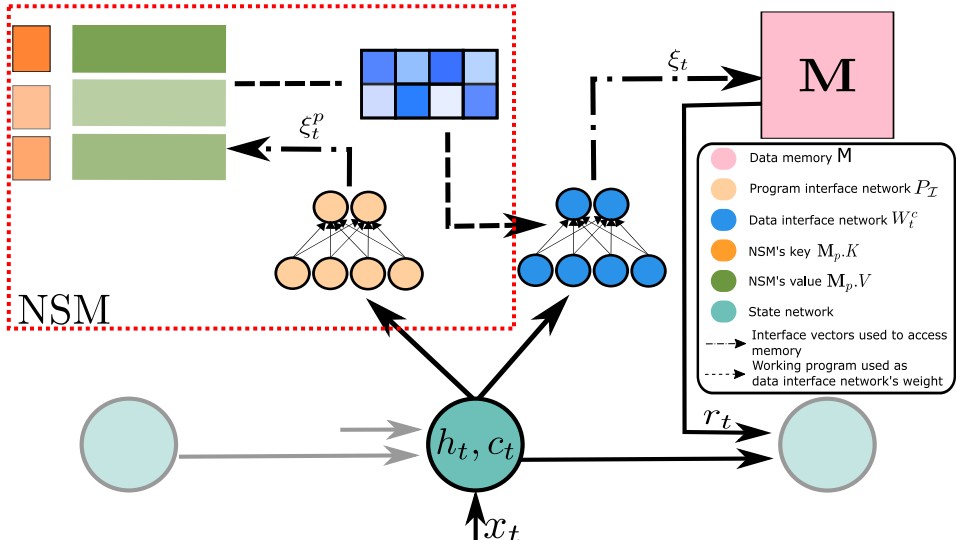

Figure 1: Introducing NSM into MANN. At each timestep, the program interface network $(P_\mathcal{I})$ receives input from the state network and queries the program memory $\mathbf{M}_p$, acquiring the working weight for the interface network $(W_t^c)$. The interface network then operates on the data memory $\mathbf{M}$.

$k_t^p$ with $\mathbf{M}_p(i).k$. That is, only the meta-network learns the mapping from context $c_t$ to program. When it falls into some local-minima (generating suboptimal $w_t^p$), the meta-network struggles to escape. In our proposal, together with the meta-network, the memory keys are learnable. When the memory keys are slowly updated, the meta-network will shift its query key generation to match the new memory keys and possibly escape from the local-minima.

For the case of multi-head NTM, we implement one NSM per control head and name this model Neural Universal Turing Machine (NUTM). One NSM per head is to ensure programs for one head do not interfere with other heads and thus, encourage functionality separation amongst heads. Each control head will read from (for read head) or write to (for write head) the data memory $\mathbf{M}$ via $memory(\xi_t, \mathbf{M})$ as described in Graves et al. (2014). It should be noted that using multiple heads is unlike using multiple controllers per head. The former increases the number of accesses to the data memory at each timestep and employs a fixed controller to compute multiple heads, which may improve capacity yet does not enable adaptability. On the contrary, the latter varies the property of each memory access across timesteps by switching the controllers and thus potential for adaptation.

Other MANNs such as DNC (Graves et al., 2016) and LRUA (Santoro et al., 2016) can be armed with NSM in this manner. We also employ the regularization loss $l_p$ to prevent the programs from collapsing, resulting in a final loss as follows,

$$Loss = Loss_{pred} + \eta_t l_p \tag{7}$$

where $Loss_{pred}$ is the prediction loss and $\eta_t$ is annealing factor, reducing as the training step increases. The details of NUTM operations are presented in Algorithm 1.

### 3.3 On the Benefit of NSM to MANN: An Explanation from Multilevel Modeling

Learning to access memory is a multi-dimensional regression problem. Given the input $c_t$, which is derived from the state $h_t$ of the controller, the aim is to generate a correct interface vector $\xi_t$ via optimizing the interface network. Instead of searching for one transformation that maps the whole space of $c_t$ to the optimal space of $\xi_t$, NSM first partitions the space of $c_t$ into subspaces, then finds multiple transformations, each of which covers subspace of

---

**Algorithm 1** Neural Universal Turing Machine

---

**Require:** a sequence $x = \{x_t\}_{t=1}^T$, a data memory $\mathbf{M}$ and $R$ program memories $\{\mathbf{M}_{p,n}\}_{n=1}^R$ corresponding to $R$ control heads
1: Initilize $h_0$, $r_0$
2: **for** $t = 1, T$ **do**
3:     $h_t, c_t = RNN([x_t, r_{t-1}], h_{t-1})$             $\triangleright$ $RNN$ can be replaced by GRU/LSTM
4:     **for** $n = 1, R$ **do**
5:         Compute the program interface $\xi_{t,n}^p \leftarrow P_{\mathcal{I},n}(c_t)$
6:         Compute the program $W_{t,n}^c \leftarrow NSM\left(\xi_{t,n}^p, \mathbf{M}_{p,n}\right)$
7:         Compute the data interface $\xi_{t,n} \leftarrow c_t W_{t,n}^c$
8:         Read $r_{t,n}$ from memory $\mathbf{M}$ (if read head) or update memory $\mathbf{M}$ (if write head) using $memory_n(\xi_{t,n}, \mathbf{M})$
9:     **end for**
10:    $r_t \leftarrow [r_{t,1}, ..., r_{t,R}]$
11: **end for**

---

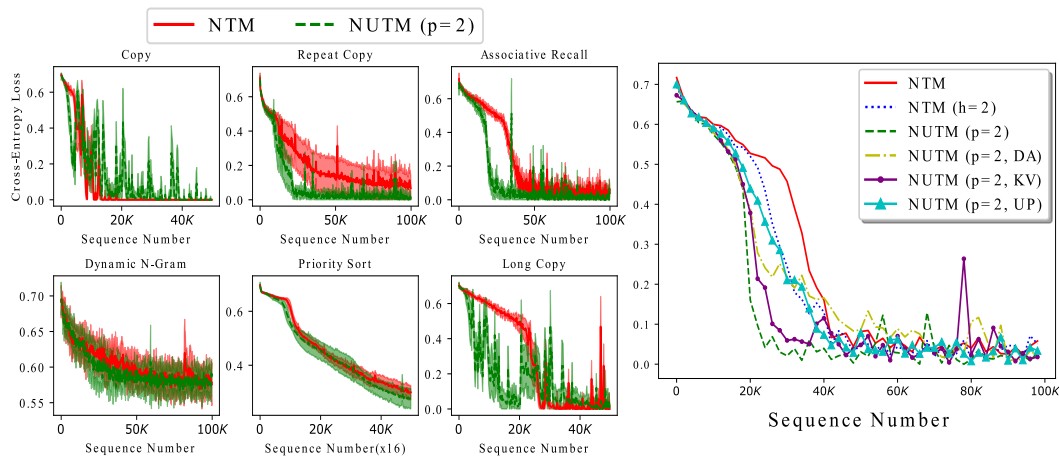

(a) Learning curves of NTM tasks        (b) Ablation study on AR

Figure 2: Learning curves on NTM tasks (a) and Associative Recall (AR) ablation study (b). Only mean is plotted in (b) for better visualization.

$c_t$. The program interface network $P_{\mathcal{I}}$ is a meta learner that routes $c_t$ to the appropriate transformation, which then maps $c_t$ to the $\xi_t$ space. This is analogous to multilevel regression in statistics (Andrew Gelman, 2006). Practical studies have shown that multilevel regression is better than ordinary regression if the input is clustered (Cohen et al., 2014; Huang, 2018).

RNNs have the capacity to learn to perform finite state computations (Casey, 1996; Tiňo et al., 1998). The states of a RNN must be grouped into partitions representing the states of the generating automaton. As Turing Machines are finite state automata augmented with an external memory tape, we expect MANN, if learnt well, will organize its state space clustered in a way to reflect the states of the emulated Turing Machine. That is, $h_t$ as well as $c_t$ should be clustered. We realize that NSM helps NTM learn better clusterization over this space (see App. A), thereby improving NTM's performances.

## 4 RESULTS

### 4.1 NTM SINGLE TASKS

In this section, we investigate the performance of NUTM on algorithmic tasks introduced in Graves et al. (2014) : Copy, Repeat Copy, Associative Recall, Dynamic N-Grams and Priority Sort. Besides these five NTM tasks, we add another task named Long Copy which

| Task | Copy | R. Copy | A. Recall | D. N-grams | P. Sort | L. Copy |
|------|------|---------|-----------|------------|---------|---------|
| NTM | **0.00** | 405.10 | 7.66 | 132.59 | 24.41 | 16.04 |
| NUTM (p=2) | **0.00** | **366.69** | **1.35** | **127.68** | **20.00** | **0.02** |

Table 1: Generalization performance of best models measured in average bit error per sequence (lower is better). For each task, we pick 1,000 longer sequences as test data.

doubles the length of training sequences in the Copy task. In these tasks, the model will be fed a sequence of input items and is required to infer a sequence of output items. Each item is represented by a binary vector.

In the experiment, we compare two models: NTM[2] and NUTM with two programs. Although the tasks are atomic, we argue that there should be at least two memory manipulation schemes across timesteps, one for encoding the inputs to the memory and another for decoding the output from the memory. The two models are trained with cross-entropy objective function under the same setting as in Graves et al. (2014) . For fair comparison, the controller hidden dimension of NUTM is set smaller to make the total number of parameters of NUTM equivalent to that of NTM. The number of memory heads for both models are always equal and set to the same value as in the original paper (details in App. C).

We run each experiments five times and report the mean with error bars of training losses for NTM tasks in Fig. 2 (a). Except for the Copy task, which is too simple, other tasks observe convergence speed improvement of NUTM over that of NTM, thereby validating the benefit of using two programs across timesteps even for the single task setting. NUTM requires fewer training samples to converge and it generalizes better to unseen sequences that are longer than training sequences. Table 1 reports the test results of the best models chosen after five runs and confirms the outperformance of NUTM over NTM for generalization.

To illustrate the program usage, we plot NUTM's program distributions across timesteps for Repeat Copy and Priority Sort in Fig. 3 (a) and (b), respectively. Examining the read head for Repeat Copy, we observe two program usage patterns corresponding to the encoding and decoding phases. As there is no reading in encoding, NUTM assigns the "no-read" strategy mainly to the "orange program". In decoding, the sequential reading is mostly done by the "blue program" with some contributions from the "orange program" when resetting reading head. Similar behaviors can be found in the write head for Priority Sort. While the encoding "fitting writing" (see Graves et al. (2014) for explanation on the strategy) is often executed by the "blue program", the decoding writing is completely taken by the "orange" program (more visualizations in App. B).

## 4.2 ABLATION STUDY ON ASSOCIATIVE RECALL

In this section, we conduct an ablation study on Associative Recall (AR) to validate the benefit of proposed components that constitute NSM. We run the task with three additional baselines: NUTM using direct attention (DA), NUTM using key-value without regularization (KV), NUTM using fixed, uniform program distribution (UP) and a vanilla NTM with 2 memory heads ($h = 2$). The meta-network $P_{\mathcal{I}}$ in DA generates the attention weight $w_t^p$ directly. The KV employs key-value attention yet excludes the regularization loss presented in Eq. (6). The training curves over 5 runs are plotted in Fig. 2 (b). The results demonstrate that DA exhibits fast yet shallow convergence. It tends to fall into local minima, which finally fails to reach zero loss. Key-value attention helps NUTM converge completely with fewer iterations. The performance is further improved with the proposed regularization loss. UP underperforms NUTM as it lacks dynamic programs. The NTM with 2 heads shows slightly better convergence compared to the NTM, yet obviously underperforms NUTM ($p = 2$) with 1 head and fewer parameters. This validates our argument on the difference between using multiple heads and multiple programs (Sec. 3.2).

---

[2]For algorithmic tasks, we choose NTM as the only baseline as NTM is known to perform and generalize well on these tasks. If NSM can help NTM in these tasks, it will probably help other MANNs as well.

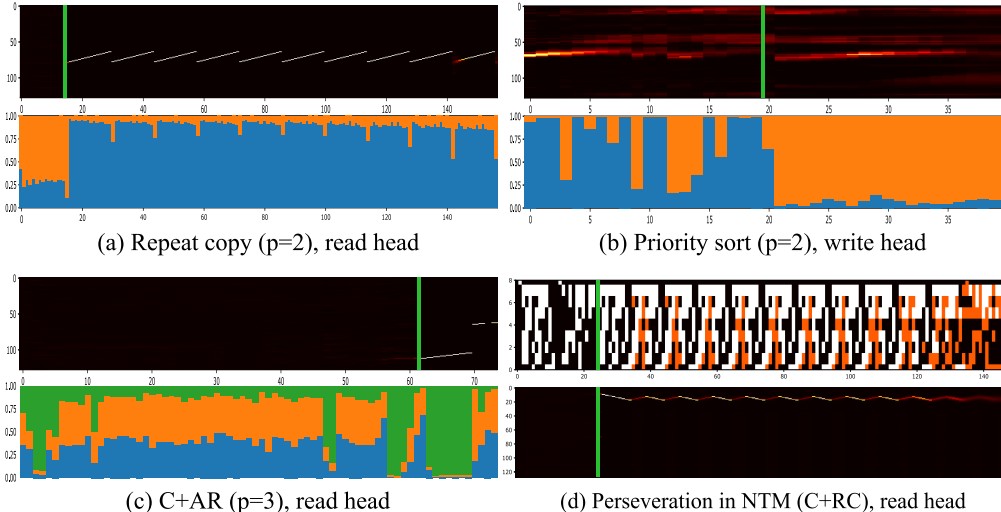

(a) Repeat copy (p=2), read head

(b) Priority sort (p=2), write head

(c) C+AR (p=3), read head

(d) Perseveration in NTM (C+RC), read head

Figure 3: (a,b,c) visualizes NUTM's executions in synthetic tasks: the upper rows are memory read (left)/write (right) locations; the lower rows are program distributions over timesteps. The green line indicates the start of the decoding phase. (d) visualizes perseveration in NTM: the upper row are input, output, predicted output with errors (orange bits); the lower row is reading location.

### 4.3 NTM Sequencing Tasks

In neuroscience, sequencing tasks test the ability to remember a series of tasks and switch tasks alternatively (Blumenfeld, 2010). A dysfunctional brain may have difficulty in changing from one task to the next and get stuck in its preferred task (perseveration phenomenon). To analyze this problem in NTM, we propose a new set of experiments in which a task is generated by sequencing a list of subtasks. The set of subtasks is chosen from the NTM single tasks (excluding Dynamic N-grams for format discrepancy) and the order of subtasks in the sequence is dictated by an indicator vector put at the beginning of the sequence. Amongst possible combinations of subtasks, we choose {Copy, Repeat Copy}(C+RC), {Copy, Associative Recall} (C+AR), {Copy, Priority Sort} (C+PS) and all (C+RC+AC+PS)[3]. The learner observes the order indicator followed by a sequence of subtasks' input items and is requested to consecutively produce the output items of each subtasks.

As shown in Fig. 4, some tasks such as Copy and Associative Recall, which are easy to solve if trained separately, become unsolvable by NTM when sequenced together. One reason is NTM fails to change the memory access behavior (perseveration). For examples, NTM keeps following repeat copy reading strategy for all timesteps in C+RC task (Fig. 3 (d)). Meanwhile, NUTM can learn to change program distribution when a new subtask appears in the sequence and thus ensure different accessing strategy per subtask (Fig. 3 (c)).

### 4.4 Continual Procedure Learning

In continual learning, catastrophic forgetting happens when a neural network quickly forgets previously acquired skills upon learning new skills (French, 1999). In this section, we prove the versatility of NSM by showing that a naive application of NSM without much modification can help NTM to mitigate catastrophic forgetting. We design an experiment similar to the Split MNIST (Zenke et al., 2017) to investigate whether NSM can improve NTM's performance. In our experiment, we let the models see the training data from the 4 tasks: Copy (C), Repeat Copy (RC), Associative Recall (AR) and Priority Sort (PS), consecutively in this order. Each task is trained in 20,000 iterations with batch size 16 (see App. C for task details). To encourage NUTM to spend exactly one program per task

---

[3]We focus on the combinations that contain Copy as Copy is the only task where NTM reach NUTM's performance. If NTM fails in these combinations, it will most likely fail in others.

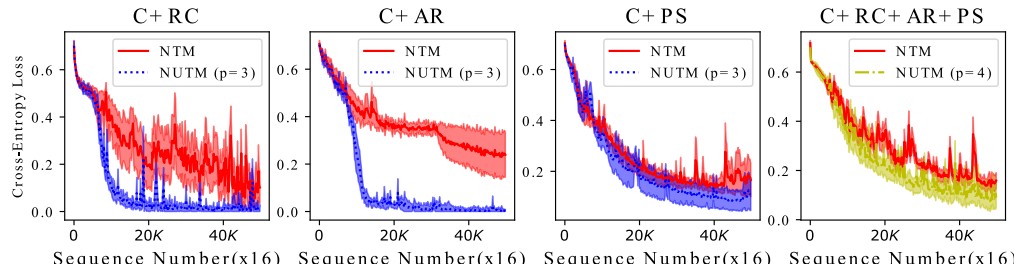

Figure 4: Learning curves on sequencing NTM tasks.

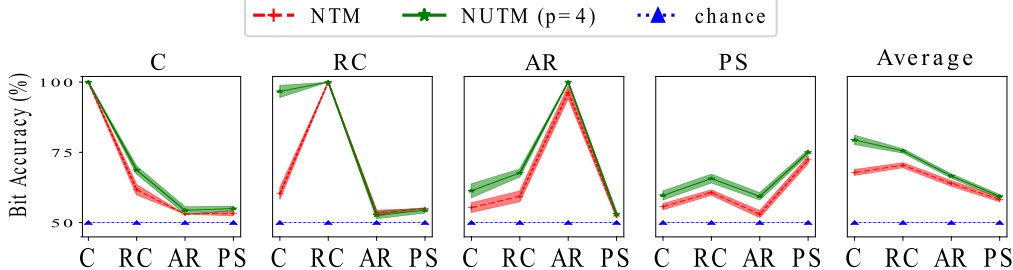

Figure 5: Mean bit accuracy with error bars for the continual algorithmic tasks. Each of the first four panels show bit accuracy on four tasks after finishing a task. The rightmost shows the average accuracy.

while freezing others, we force "hard" attention over the programs by replacing the softmax function in Eq. 5 with the Gumbel-softmax (Jang et al., 2016). Also, to ignore catastrophic forgetting in the state network, we use Feedforward controllers in the two baselines.

After finishing one task, we evaluate the bit accuracy $-$measured by $1-$(bit error per sequence/total bits per sequence) over 4 tasks. As shown in in Fig. 5, NUTM outperforms NTM by a moderate margin (10-40% per task). Although NUTM also experiences catastrophic forgetting, it somehow preserves some memories of previous tasks. Especially, NUTM keeps performing perfectly on Copy even after it learns Repeat Copy. For other dissimilar task transitions, the performance drops significantly, which requires more effort to bring NSM to continual learning.

### 4.5 Few-shot Learning

Few-shot learning or meta learning tests the ability to rapidly adapt within a task while gradually capturing the way the task structure varies (Thrun, 1998). By storing sample-class bindings, MANNs are capable of classifying new data after seeing only few samples (Santoro et al., 2016). As NSM gives flexible memory controls, it makes MANN more adaptive to changes and thus perform better in this setting. To verify that, we apply NSM to the LRUA memory and follow the experiments introduced in Santoro et al. (2016) , using the Omniglot dataset to measure few-shot classification accuracy. The dataset includes images of 1623 characters, with 20 examples of each character. During training, a sequence (episode) of images are randomly selected from $C$ classes of characters in the training set (1200 characters), where $C = 5, 10$ corresponding to sequence length of 50, 75, respectively. Each class is assigned a random label which shuffles between episodes and is revealed to the models after each prediction. After 100,000 episodes of training, the models are tested with unseen images from the testing set (423 characters). The two baselines are MANN and NUTM (both use LRUA core). For NUTM, we only tune $p$ and pick the best values: $p = 2$ and $p = 3$ for 5 classes and 10 classes, respectively.

Table 11 reports the classification accuracy when the models see characters for the second, third and fifth time. NUTM generally achieves better results than MANN, especially when the number of classes increases, demanding more adaptation within an episode. For the

| Model | Persistent memory[5] | 5 classes | | | 10 classes | | |
|---|---|---|---|---|---|---|---|
| | | $2^{nd}$ | $3^{rd}$ | $5^{th}$ | $2^{nd}$ | $3^{rd}$ | $5^{th}$ |
| MANN (LRUA)* | No | 82.8 | 91.0 | 94.9 | - | - | - |
| MANN (LRUA) | No | 82.3 | 88.7 | 92.3 | 52.7 | 60.6 | 64.7 |
| NUTM (LRUA) | No | **85.7** | **91.3** | **95.5** | **68.0** | **78.1** | **82.8** |
| MANN (LRUA) | Yes | 66.2 | 73.4 | 81.0 | 51.3 | 59.2 | 63.3 |
| NUTM (LRUA) | Yes | **77.8** | **85.8** | **89.8** | **69.0** | **77.9** | **82.7** |

Table 2: Test-set classification accuracy (%) on the Omniglot dataset after 100,000 episodes of training. * denotes available results from (Santoro et al., 2016).

| Model | Error |
|---|---|
| DNC(Graves et al., 2016) | $16.7 \pm 7.6$ |
| SDNC(Rae et al., 2016) | $6.4 \pm 2.5$ |
| ADNC(Franke et al., 2018) | $6.3 \pm 2.7$ |
| DNC-MD(Csordas & Schmidhuber, 2019) | $9.5 \pm 1.6$ |
| NUTM (DNC core, p=1) | $9.7 \pm 3.5$ |
| NUTM (DNC core, p=2) | $7.5 \pm 1.6$ |
| NUTM (DNC core, p=4) | $\mathbf{5.6 \pm 1.9}$ |

Table 3: Mean and s.d. for bAbI error (%).

persistent memory mode, which demands fast forgetting old experiences in previous episodes, NUTM outperforms MANN significantly (10-20%)[4]. Readers are referred to App. D for more details on learning curves and more results of the models.

## 4.6 Text Question Answering

Reading comprehension typically involves an iterative process of multiple actions such as reading the story, reading the question, outputting the answers and other implicit reasoning steps (Weston et al., 2015). We apply NUTM to the question answering domain by replacing the NTM core with DNC (Graves et al., 2016). Compared to NTM's sequential addressing, dynamic memory addressing in DNC is more powerful and thus suitable for NSM integration to solve non-algorithmic problems such as question answering. Following previous works of DNC, we use bAbI dataset (Weston et al., 2015) to measure the performance of the NUTM with DNC core (three variants $p = 1$, $p = 2$ and $p = 4$). In the dataset, each story is followed by a series of questions and the network reads all word by word, then predicts the answers. Although synthetically generated, bAbI is a good benchmark that tests 20 aspects of natural language reasoning including complex skills such as induction and counting,

We found that increasing number of programs helps NUTM improve performance. In particular, NUTM with 4 programs, after 50 epochs jointly trained on all 20 question types, can achieve a mean test error rate of 3.3% and manages to solve 19/20 tasks (a task is considered solved if its error <5%). The mean and s.d. across 10 runs are also compared with other results reported by recent works (see Table 3). Excluding baselines under different setups, our result is the best reported mean result on bAbI that we are aware of. More details are described in App. E.

---

[4]It should be noted that our goal was not to achieve state of the art performance on this dataset. It was to exhibit the benefit of NSM to MANN. Compared to current methods, the MANN and NUTM used in our experiments do not use CNN to extract visual features, thus achieve lower accuracy than recent state-of-the-arts.

[5]If the memory is not artificially erased between episodes, it is called persistent. This mode is hard for the case of 5 classes as shown in (Santoro et al., 2016)

## 5    Related Work

Previous investigations into MANNs mostly revolve around memory access mechanisms. The works in Graves et al. (2014; 2016) introduce content-based, location-based and dynamic memory reading/writing. Further, Rae et al. (2016) scales to bigger memory by sparse access; Le et al. (2019) optimizes memory operations with uniform writing; and MANNs with extra memory have been proposed (Le et al., 2018b). However, these works keep using memory for storing data rather than the weights of the network and thus parallel to our approach. Other DNC modifications (Csordas & Schmidhuber, 2019; Franke et al., 2018) are also orthogonal to our work.

Another line of related work involves modularization of neural networks, which is designed for visual question answering. In module networks (Andreas et al., 2016b;a), the modules are manually aligned with predefined concepts and the order of execution is decided by the question. Although the module in these works resembles the program in NSM, our model is more generic and flexible with soft-attention over programs and thus fully differentiable. Further, the motivation of NSM does not limit to a specific application. Rather, NSM aims to help MANN reach general-purpose computability.

If we view NSM network as a dynamic weight generator, the program in NSM can be linked to fast weight (von der Malsburg, 1981; Hinton & Plaut, 1987; Schmidhuber, 1993b). These papers share the idea of using different weights across timesteps to enable dynamic adaptation. Using outer-product is a common way to implement fast-weight (Schmidhuber, 1993a; Ba et al., 2016; Schlag & Schmidhuber, 2017). These fast weights are directly generated and thus different from our programs, which are interpolated from a set of slow weights.

Tensor/Multiplicative RNN (Sutskever et al., 2011) and Hypernetwork (Ha et al., 2016) are also relevant related works. These methods attempt to make the working weight of RNNs dependent on the input to enable quick adaption through time. Nevertheless, they do not support modularity. In particular, Hypernetwork generates scaling factors for the single weight of the main RNN. It does not aim to use multiple slow-weights (programs) and thus, different from our approach. Tensor RNN is closer to our idea when the authors propose to store $M$ slow-weights, where $M$ is the number of input dimension, which is acknowledged impractical. Unlike our approach, they do not use a meta-network to generate convex combinations amongst weights. Instead, they propose Multiplicative RNN that factorizes the working weight to product of three matrices, which looses modularity. On the contrary, we explicitly model the working weight as an interpolation of multiple programs and use a meta-network to generate the coefficients. This design facilitates modularity because each program is trained towards some functionality and can be switched or combined with each other to perform the current task. Last but not least, while the related works focus on improving RNN with fast-weight, we aim to reach a neural simulation of Universal Turing Machine, in which fast-weight is a way to implement stored-program principle.

## 6    Conclusions

This paper introduces the Neural Stored-program Memory (NSM), a new type of external memory for neural networks. The memory, which takes inspirations from the stored-program memory in computer architecture, gives memory-augmented neural networks (MANNs) flexibility to change their control programs through time while maintaining differentiability. The mechanism simulates modern computer behavior, potential making MANNs truly neural computers. Our experiments demonstrated that when coupled with our model, the Neural Turing Machine learns algorithms better and adapts faster to new tasks at both sequence and sample levels. When used in few-shot learning, our method helps MANN as well. We also applied the NSM to the Differentiable Neural Computer and observed a significant improvement, reaching the state-of-the-arts in the bAbI task. Although this paper limits to MANN integration, other neural networks can also reap benefits from our proposed model, which will be explored in future works.

ACKNOWLEDGMENTS

This research was partially funded by the Australian Government through the Australian Research Council (ARC). Prof Venkatesh is the recipient of an ARC Australian Laureate Fellowship (FL170100006).

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

## APPENDIX

## A    CLUSTERING ON THE LATENT SPACE

As previously mentioned in Sec. 3.3, MANN should let its states form clusters to well-simulate Turing Machine. Fig. 6 (a) and (c) show NTM actually organizes its $c_t$ space into clusters corresponding to processing states (e.g, encoding and decoding). NUTM, which explicitly partitions this space, clearly learn better clusters of $c_t$ (see Fig. 6 (b) and (d)). This contributes to NUTM's outperformance over NTM.

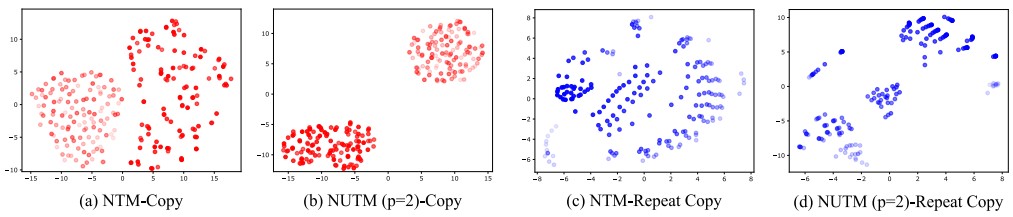

(a) NTM-Copy     (b) NUTM (p=2)-Copy     (c) NTM-Repeat Copy     (d) NUTM (p=2)-Repeat Copy

Figure 6: Visualization of the first two principal components of $c_t$ space in NTM (a,c) and NUTM (b,d) for Copy (red) and Repeat Copy (blue). Fader color denotes lower timestep in a sequence. Both can learn clusters of hidden states yet NUTM exhibits clearer partition.

## B    PROGRAM USAGE VISUALIZATIONS

B.1 and B.2 visualize the best inferences of NUTM on test data from single and sequencing tasks. Each plot starts with the input sequence and the predicted output sequence with error bits in the first row[6]. The second and fourth rows depict the read and write locations on data memory, respectively. The third and fifth rows depict the program distribution of the read head and write head, respectively. B.3 visualizes random failed predictions of NTM on sequencing tasks. The plots follow previous pattern except for the program distribution rows.

---

[6]Normally, black is bit 0, white is bit 1 in vector data. Orange is prediction error. In tasks including priority sort, because data vectors not only include value 0-1, but also other float values (e.g., priority score), the color scale is automatically changed. Basically, error bit is given darker color than 0 and lighter color than 1. For example, in priority sort task, yellow is prediction error, and orange is bit 1.

## B.1 VISUALIZATION ON PROGRAM DISTRIBUTION ACROSS TIMESTEPS (SINGLE TASKS)

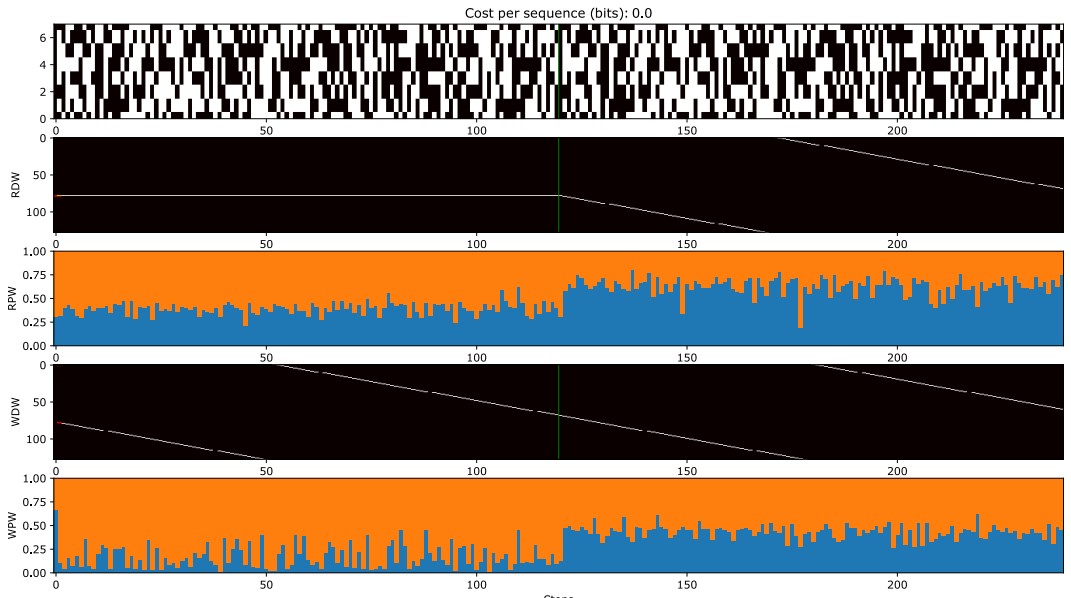

Figure 7: Copy (p=2).

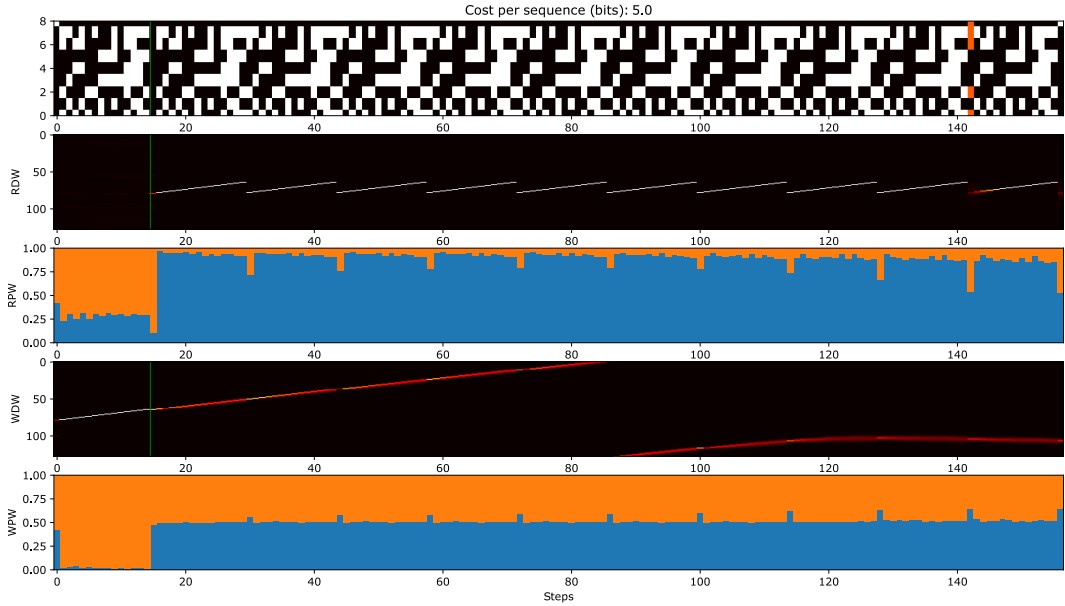

Figure 8: Repeat Copy (p=2).

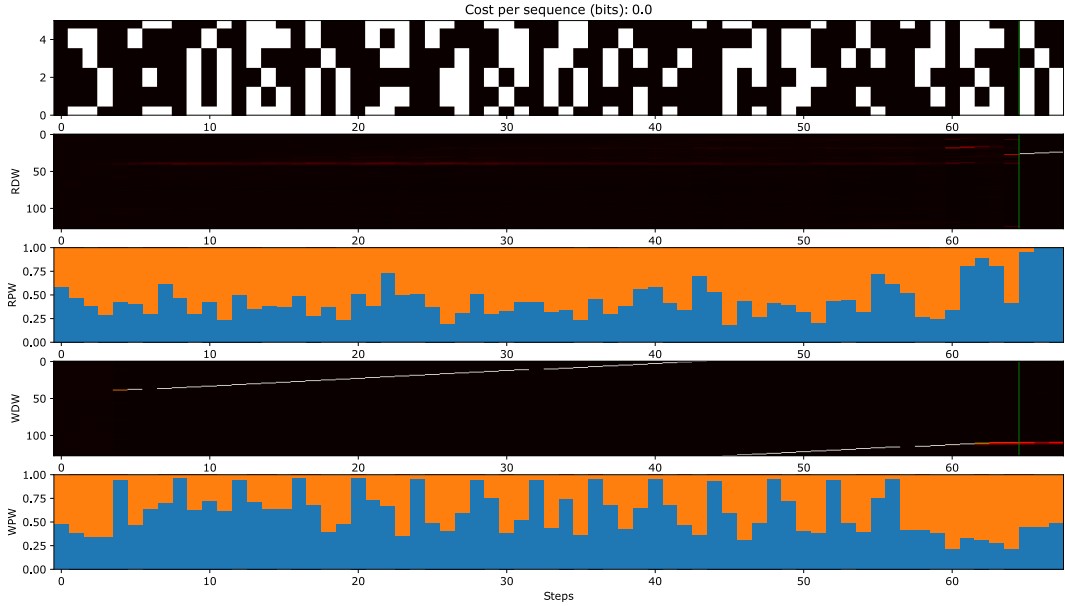

Figure 9: Associative Recall (p=2).

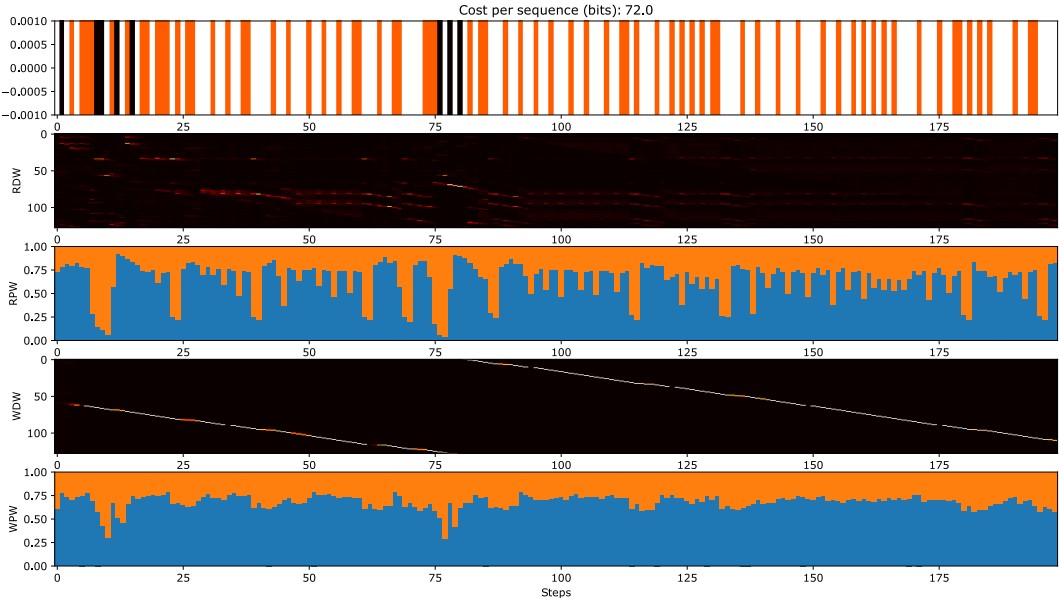

Figure 10: Dynamic N-grams (p=2).

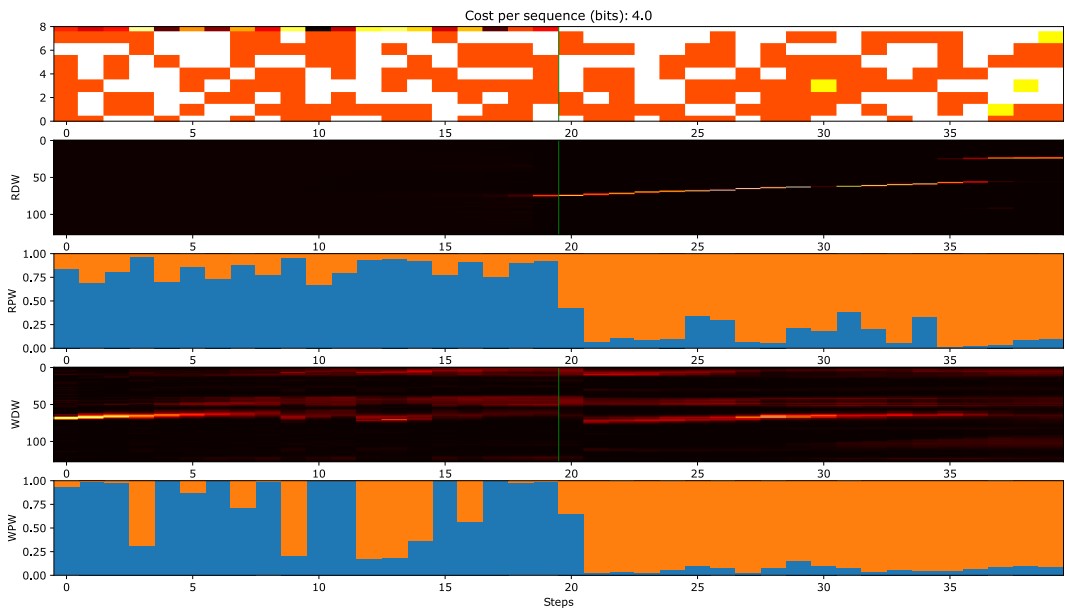

Figure 11: Priority Sort (p=2).

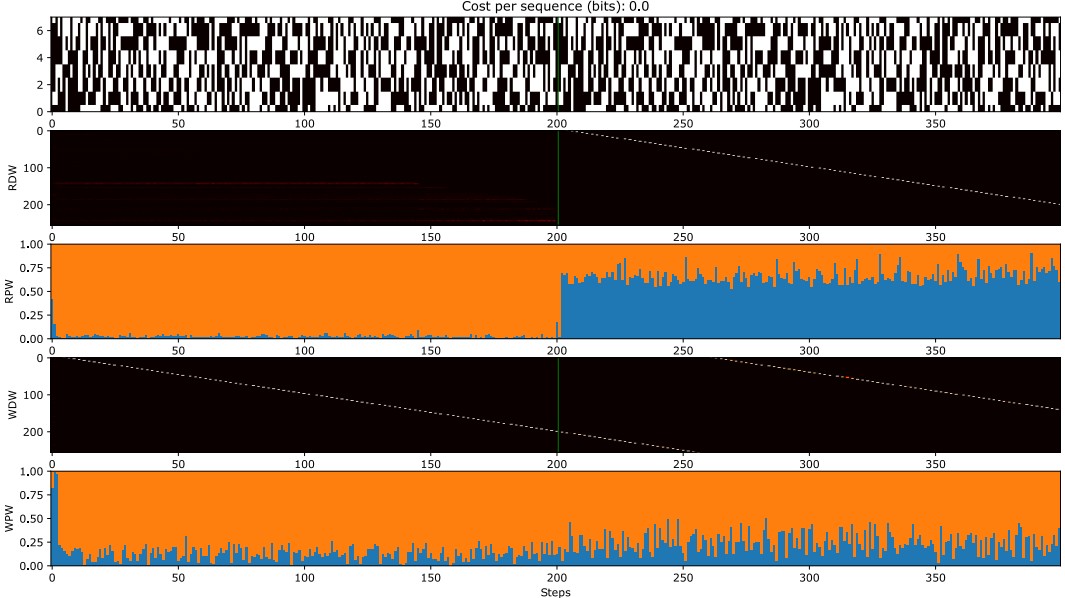

Figure 12: Long Copy (p=2).

## B.2 Visualization on program distribution across timesteps (sequencing tasks)

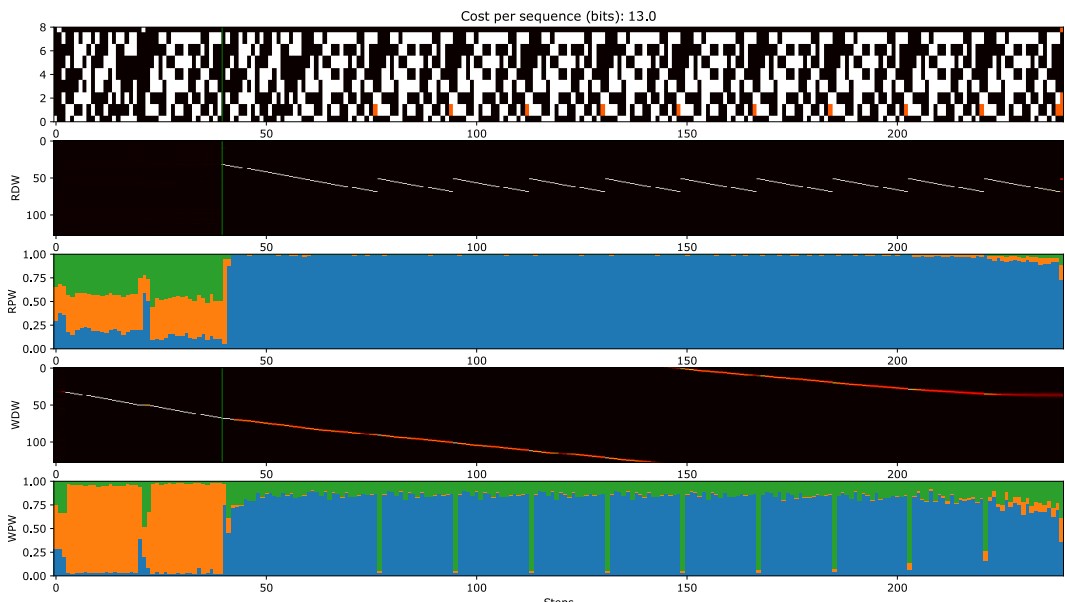

Figure 13: Copy+Repeat Copy (p=3).

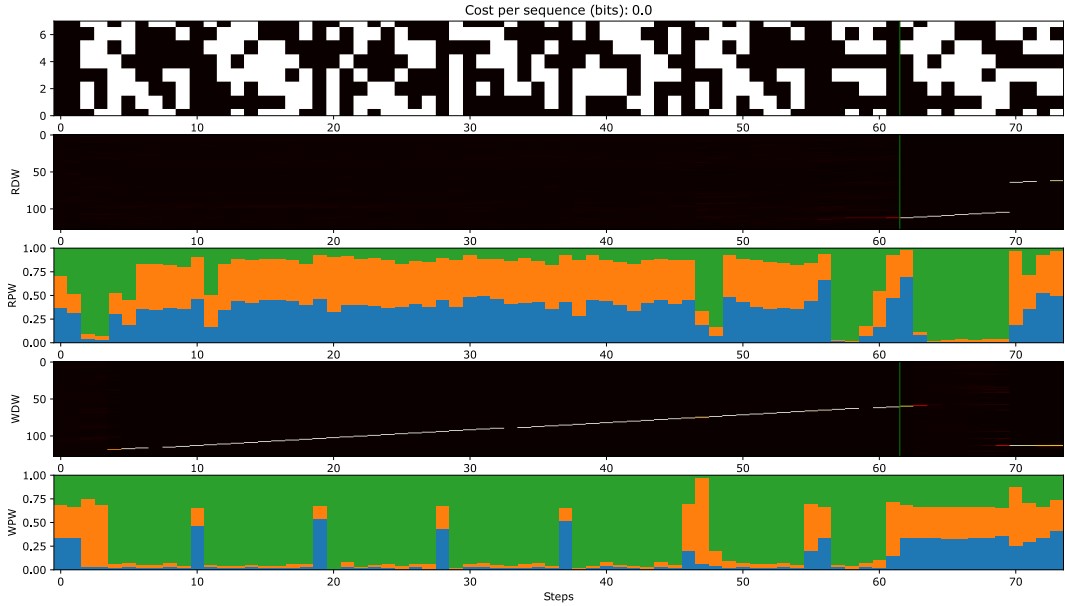

Figure 14: Copy+Associative Recall (p=3).

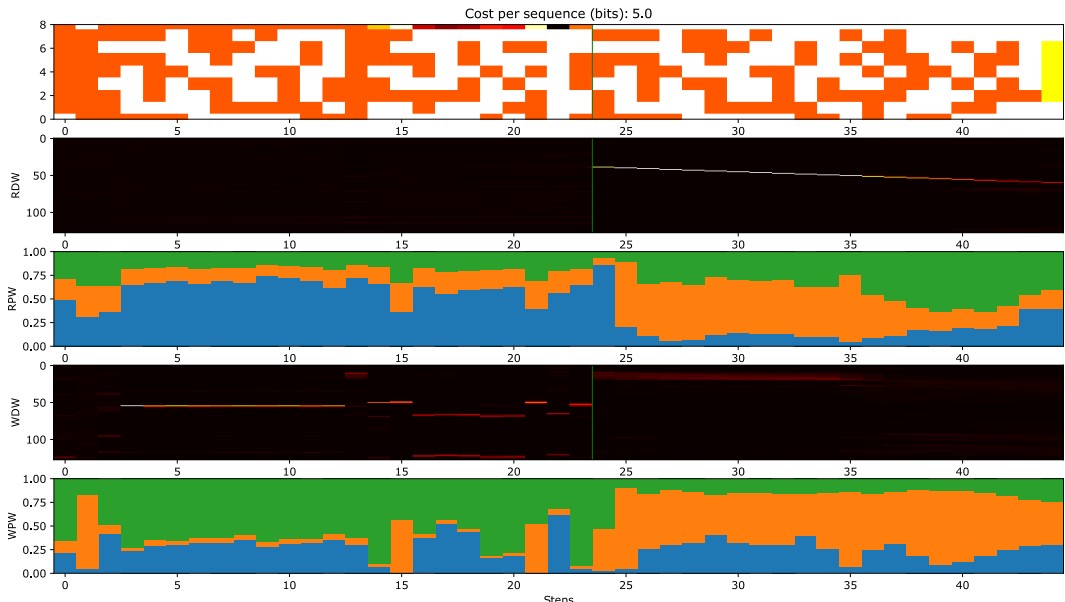

Figure 15: Copy+Priority Sort (p=3).

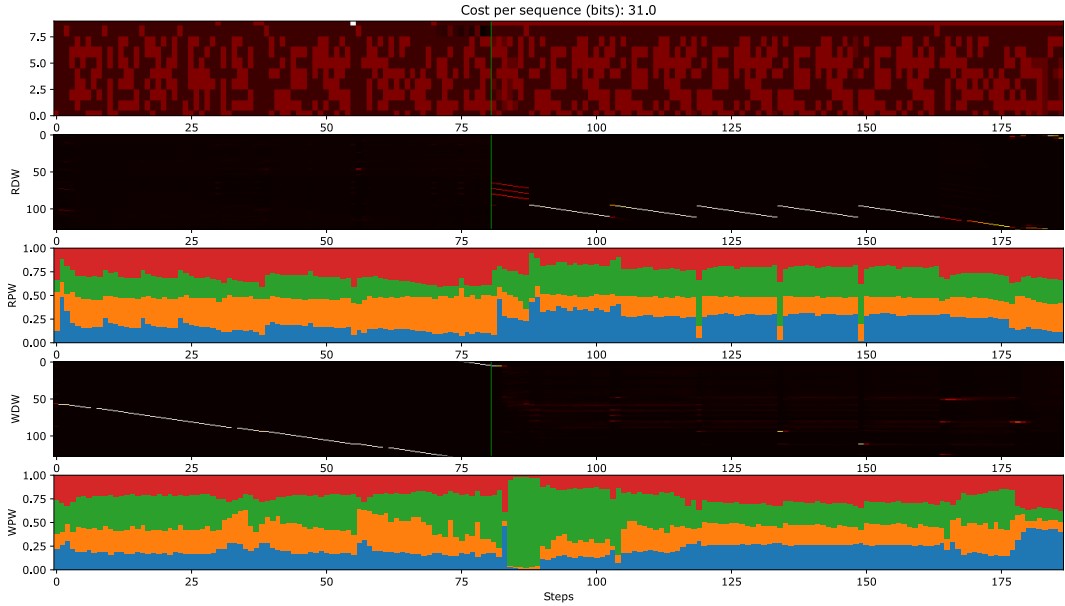

Figure 16: Copy+Repeat Copy+Associative Recall+Priority Sort (p=4).

## B.3 Perseveration phenomenon in NTM (sequencing tasks)

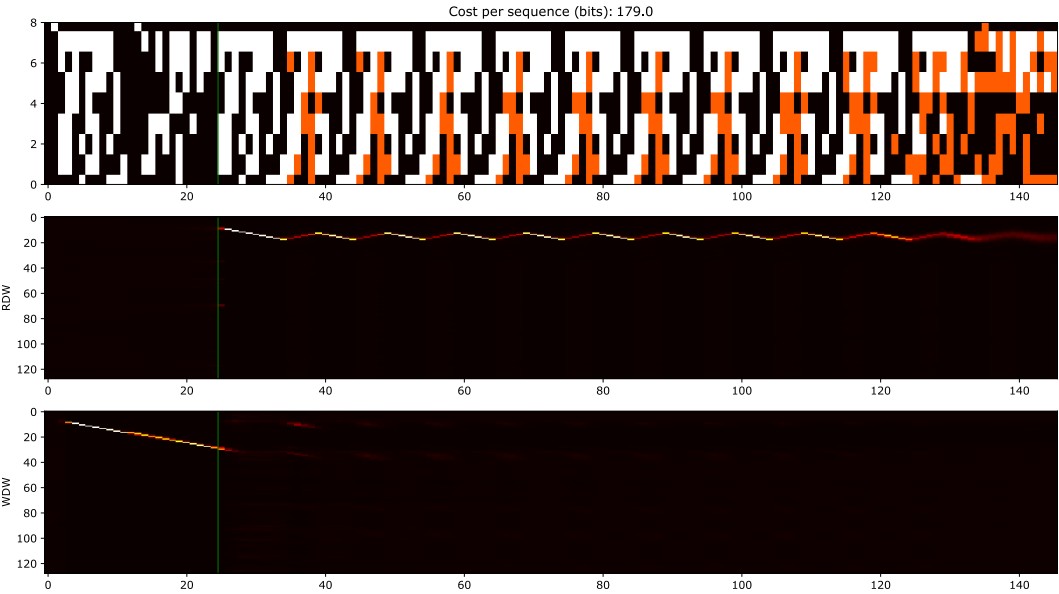

Figure 17: Copy+Repeat Copy perseveration (only Repeat Copy).

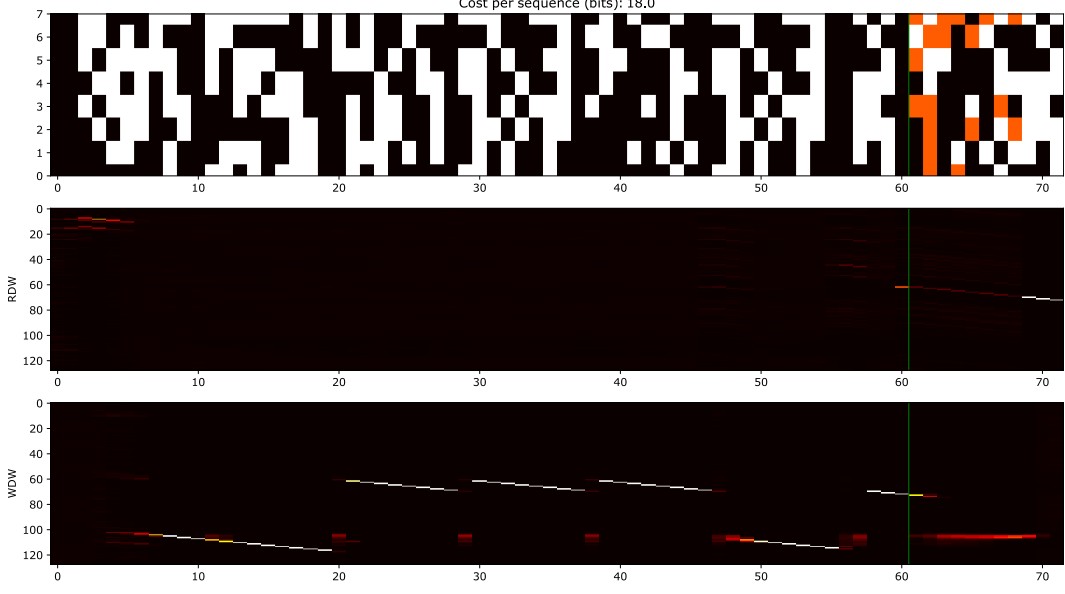

Figure 18: Copy+Associative Recall perseveration (only Copy).

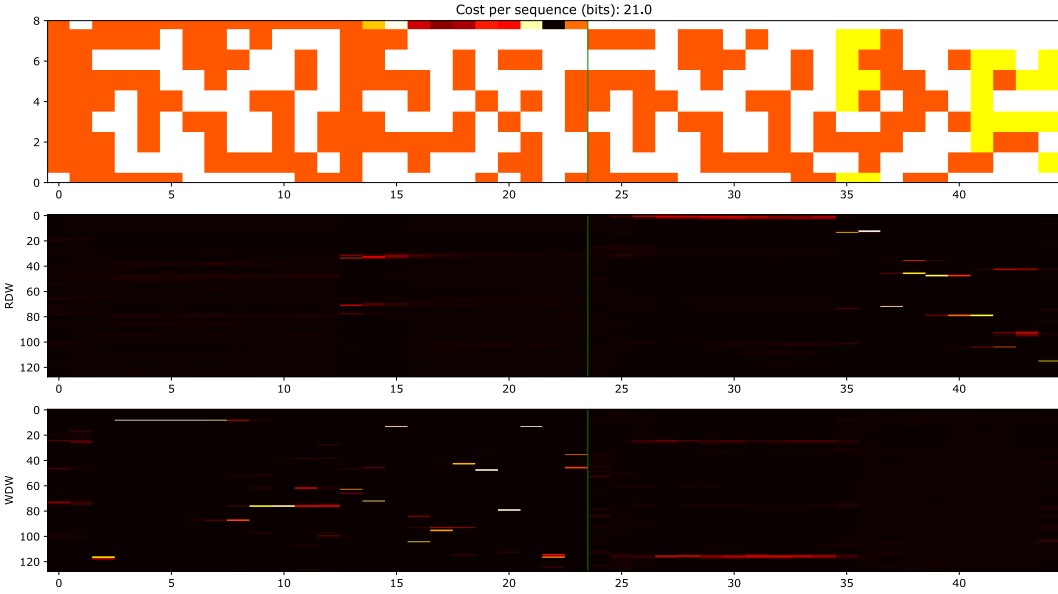

Figure 19: Copy+Priority Sort perseveration (only Copy).

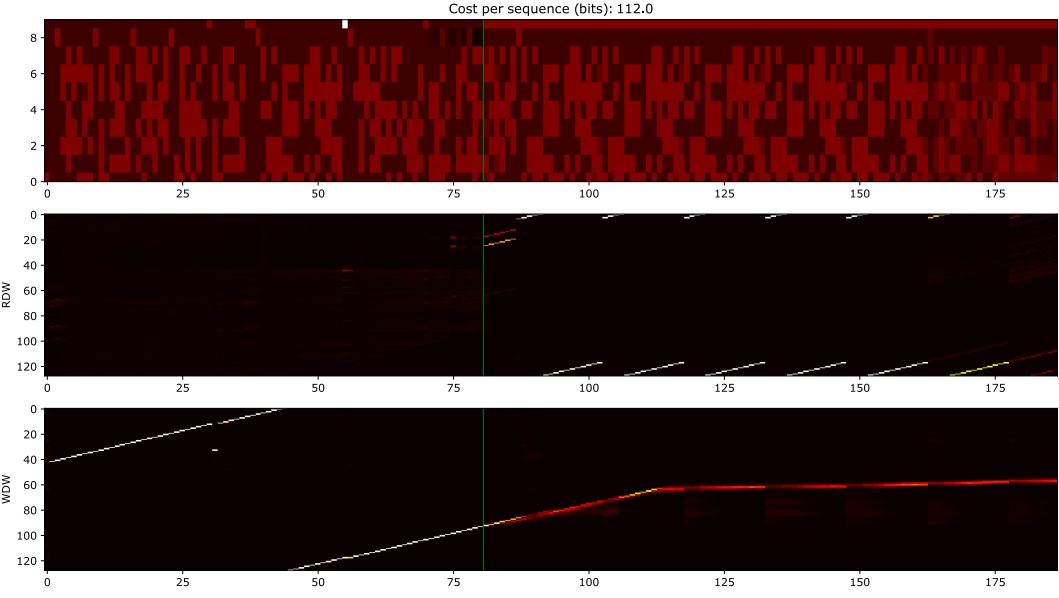

Figure 20: Copy+Repeat Copy+Associative Recall+Priority Sort perseveration (only Repeat Copy).

## C    Details on Synthetic Tasks

### C.1    NTM single tasks

| Tasks | #Read/Write Head[7] | | Controller Size | | Memory Size | | #Parameters | |
|---|---|---|---|---|---|---|---|---|
| | NTM | NUTM | NTM | NUTM | NTM | NUTM | NTM | NUTM |
| Copy | 1 | 1 | 100 | 80 | 128 | 128 | 63,260 | 52,206 |
| Repeat Copy | 1 | 1 | 100 | 80 | 128 | 128 | 63,381 | 52,307 |
| Associative Recall | 1 | 1 | 100 | 80 | 128 | 128 | 62,218 | 51,364 |
| Dynamic N-grams | 1 | 1 | 100 | 80 | 128 | 128 | 58,813 | 48,619 |
| Priority Sort | 5 | 5 | 200 | 150 | 128 | 128 | 344,068 | 302,398 |
| Long Copy | 1 | 1 | 100 | 80 | 256 | 256 | 63,260 | 52,206 |

Table 4: Model hyper-parameters (single tasks).

| Tasks | Training | Testing |
|---|---|---|
| Copy | Sequence length range: [1, 20] | Sequence length: 120 |
| Repeat Copy | Sequence length range: [1, 10] #Repeat range: [1, 10] | Sequence length range: [10, 20] #Repeat range: [10, 20] |
| Associative Recall | #Item range: [2, 6] Item length: 3 | #Item range: [6, 20] Item length: 3 |
| Dynamic N-grams | Sequence length: 50 | Sequence length: 200 |
| Priority Sort | #Item: 20 #Sorted Item: 16 | #Item: 20 #Sorted Item: 20 |
| Long Copy | Sequence length range: [1, 40] | Sequence length: 200 |

Table 5: Task settings (single tasks).

### C.2    NTM sequencing tasks

| Tasks | #Read/Write Head | | Controller Size | | Memory Size | | #Parameters | |
|---|---|---|---|---|---|---|---|---|
| | NTM | NUTM | NTM | NUTM | NTM | NUTM | NTM | NUTM |
| C+RC | 1 | 1 | 200 | 150 | 128 | 128 | 206,481 | 153,941 |
| C+AR | 1 | 1 | 200 | 150 | 128 | 128 | 206,260 | 153,770 |
| C+PS | 3 | 3 | 200 | 150 | 128 | 128 | 275,564 | 263,894 |
| C+RC+AR+PS | 3 | 3 | 250 | 200 | 128 | 128 | 394,575 | 448,379 |

Table 6: Model hyper-parameters (sequencing tasks).

---

[7]In NTM, the number of read and write heads are equal.

| Tasks | Training | Testing |
|---|---|---|
| C+RC | Sequence length range: [1, 10]
#Repeat range: [1, 10] | Sequence length range: [10, 20]
#Repeat range: [10, 15] |
| C+AR | Sequence length range: [1, 10]
#Item range: [2, 4]
Item length: 8 | Sequence length range: [10, 20]
#Item range: [4, 6]
Item length: 8 |
| C+PS | Sequence length range: [1, 10]
#Item: 10
#Sorted Item: 8 | Sequence length range: [10, 20]
#Item: 10
#Sorted Item: 10 |
| C+RC+AR+PS | Sequence length range: [1, 10]
#Repeat range: [1, 5]
#Item range: [2, 4]
Item length: 6
#Item: 10
#Sorted Item: 8 | Sequence length range: [10, 20]
#Repeat: 6
#Item: 5
Item length: 6
#Item: 10
#Sorted Item: 10 |

Table 7: Task settings (sequencing tasks).

## C.3 CONTINUAL PROCEDURE LEARNING TASKS

| #Read/Write Head | | Controller Size | | Memory Size | | #Parameters | |
|---|---|---|---|---|---|---|---|
| NTM | NUTM | NTM | NUTM | NTM | NUTM | NTM | NUTM |
| 1 | 1 | 200 | 150 | 128 | 128 | 206,444 | 196,590 |

Table 8: Model hyper-parameters (continual procedure learning tasks). NUTM uses 6 programs per head.

| Tasks | Training | Testing |
|---|---|---|
| Copy | Sequence length range: [1, 10] | Sequence length range: [1, 10] |
| Repeat Copy | Sequence length range: [1, 5]
#Repeat range: [1, 5] | Sequence length range: [1, 5]
#Repeat range: [1, 5] |
| Associative Recall | Sequence length: 3
#Item range: [2, 3]
Item length: 3 | Sequence length: 3
#Item range: [2, 3]
Item length: 3 |
| Priority Sort | #Item: 10
#Sorted Item: 8 | #Item: 10
#Sorted Item: 8 |

Table 9: Task settings (continual procedure learning tasks).

## D DETAILS ON FEW-SHOT LEARNING TASK

We use similar hyper-parameters as in Santoro et al. (2016) , which are reported in Tab. 10.

| Model | $p$ | #Read Head | #Write Head | Controller Size | $N$ | $M$ | $\mathbf{M}_p.K$ Size |
|---|---|---|---|---|---|---|---|
| MANN (LRUA) | 1 | 4 | 1 | 200 | 128 | 40 | 0 |
| NUTM (LRUA) | 2 | 4 | 1 | 180 | 128 | 40 | 2 |
| NUTM (LRUA) | 3 | 4 | 1 | 150 | 128 | 40 | 3 |

Table 10: Hyper-parameters for few-shot learning. All models use RMSprop optimizer with learning rate $10^{-4}$.

Testing accuracy through time is listed below,

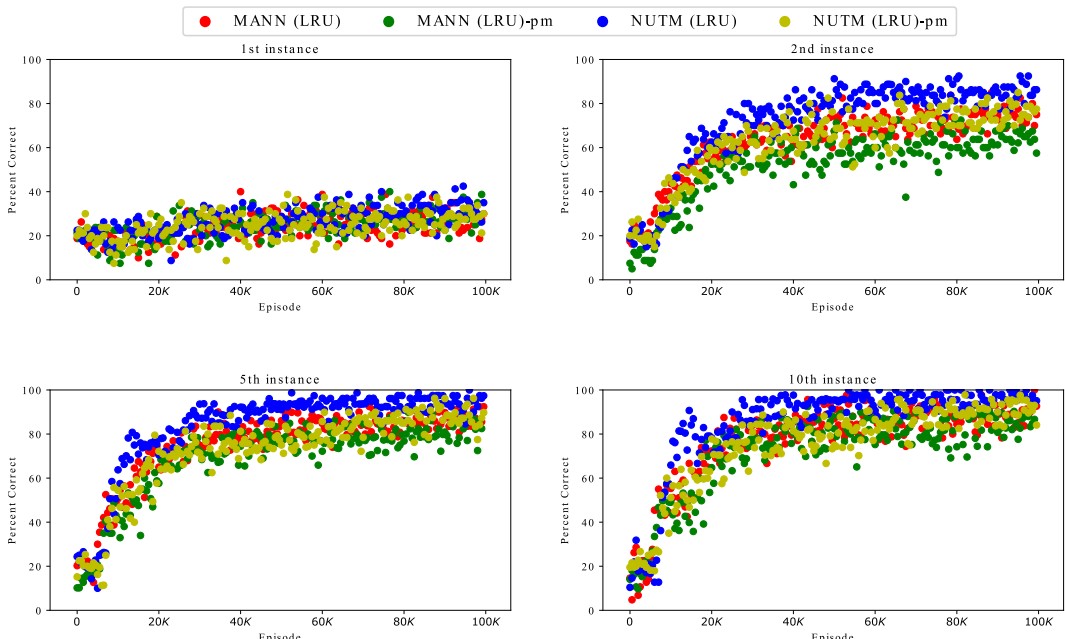

Figure 21: Testing accuracy during training (five random classes/episode, one-hot vector labels, of length 50).

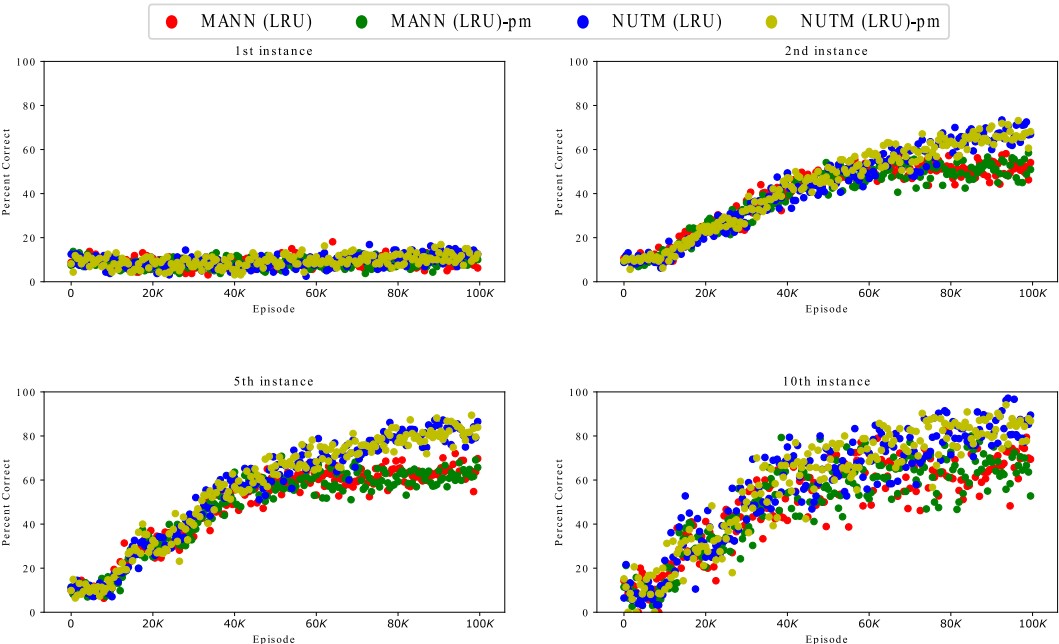

Figure 22: Testing accuracy during training (ten random classes/episode, one-hot vector labels, of length 75).

---

[8]If the memory is not artificially erased between episodes, it is called persistent. This mode is hard for the case of 5 classes as shown in Santoro et al. (2016)

| Model | Persistent memory[8] | 5 classes | | | 10 classes | | |
|---|---|---|---|---|---|---|---|
| | | $2^{nd}$ | $3^{rd}$ | $5^{th}$ | $2^{nd}$ | $3^{rd}$ | $5^{th}$ |
| MANN (LRUA)* | No | 82.8 | 91.0 | 94.9 | - | - | - |
| MANN (LRUA) | No | 82.3 | 88.7 | 92.3 | 52.7 | 60.6 | 64.7 |
| NUTM (LRUA) | No | **85.7** | **91.3** | **95.5** | **68.0** | **78.1** | **82.8** |
| Human* | Yes | 57.3 | 70.1 | 81.4 | - | - | - |
| MANN (LRUA)* | Yes | $\approx 58.0$ | - | $\approx 75.0$ | $\approx 60.0$ | - | $\approx 80.0$ |
| MANN (LRUA) | Yes | 66.2 | 73.4 | 81.0 | 51.3 | 59.2 | 63.3 |
| NUTM (LRUA) | Yes | **77.8** | **85.8** | **89.8** | **69.0** | **77.9** | **82.7** |

Table 11: Test-set classification accuracy (%) on the Omniglot dataset after 100,000 episodes of training. * denotes available results from Santoro et al. (2016) (some are estimated from plotted figures).

E  DETAILS ON BABI TASK

We train the models using RMSprop optimizer with fixed learning rate of $10^{-4}$ and momentum of 0.9. The batch size is 32 and we adopt layer normalization (Lei Ba et al., 2016) to DNC's layers. Following Franke et al. (2018) practice, we also remove temporal linkage for faster training. The details of hyper-parameters are listed in Table 12. Full NUTM ($p = 4$) results are reported in Table 13.

| #Read Head | #Write Head | Controller Size | $N$ | $M$ | $p$ | $\mathbf{M}_p.K$ Size | #Parameters |
|---|---|---|---|---|---|---|---|
| 4 | 1 | 256 | 196 | 64 | $1^9$ | 1 | 891,136 |
| 4 | 1 | 200 | 196 | 64 | 2 | 2 | 934,787 |
| 4 | 1 | 172 | 196 | 64 | 4 | 4 | 794,773 |

Table 12: NUTM hyper-parameters for bAbI.

| Task | bAbI Best Results | bAbI Mean Results |
|---|---|---|
| 1: 1 supporting fact | 0.0 | $0.0 \pm 0.0$ |
| 2: 2 supporting facts | 0.2 | $0.6 \pm 0.3$ |
| 3: 3 supporting facts | 4.0 | $7.6 \pm 3.9$ |
| 4: 2 argument relations | 0.0 | $0.0 \pm 0.0$ |
| 5: 3 argument relations | 0.4 | $1.0 \pm 0.4$ |
| 6: yes/no questions | 0.0 | $0.0 \pm 0.0$ |
| 7: counting | 1.9 | $1.5 \pm 0.8$ |
| 8: lists/sets | 0.6 | $0.3 \pm 0.2$ |
| 9: simple negation | 0.0 | $0.0 \pm 0.0$ |
| 10: indefinite knowledge | 0.1 | $0.1 \pm 0.0$ |
| 11: basic coreference | 0.0 | $0.0 \pm 0.0$ |
| 12: conjunction | 0.0 | $0.0 \pm 0.0$ |
| 13: compound coreference | 0.1 | $0.0 \pm 0.0$ |
| 14: time reasoning | 0.3 | $1.6 \pm 2.2$ |
| 15: basic deduction | 0.0 | $2.6 \pm 8.3$ |
| 16: basic induction | 49.3 | $52.0 \pm 1.7$ |
| 17: positional reasoning | 4.7 | $18.4 \pm 12.7$ |
| 18: size reasoning | 0.4 | $1.6 \pm 1.1$ |
| 19: path finding | 4.3 | $23.7 \pm 32.2$ |
| 20: agent's motivation | 0.0 | $0.0 \pm 0.0$ |
| Mean Error (%) | 3.3 | $5.6 \pm 1.9$ |
| Failed (Err. >5%) | 1 | $3 \pm 1.2$ |

Table 13: NUTM ($p = 4$) bAbI best and mean errors (%).

---

[9]When $p = 1$, the model converges to layer-normed DNC

## F   Example of memory operation function in NTM

In NTM, $\xi_t = \{\beta_t, k_t, g_t, s_t, e_t, v_t\}$. The memory addressing weight is initially computed by content-based attention,

$$w_t^c(i) = \frac{\exp\left(\beta_t m\left(k_t, M_t(i)\right)\right)}{\sum\limits_{j=1}^{D} \exp\left(\beta_t m\left(k_t, M_t(j)\right)\right)} \tag{8}$$

Here, $w_t^c \in \mathbb{R}^N$ is the content-based weight, $\beta_t$ is a strength scalar, and $m$ is implemented as cosine similarity

$$m\left(k_t, M_t(i)\right) = \frac{k_t \cdot M_t(i)}{||k_t|| \cdot ||M_t(i)||} \tag{9}$$

In addition, NTM supports location-based addressing started with an interpolation between content-based weight and the previous weight

$$w_t^g = g_t w_t^c + (1 - g_t) w_t \tag{10}$$

where $g_t$ is the interpolation gate that determines to use (or ignore) content-based addressing. Then, NTM can shift the address to other rows by performing convolution shift modulo $R$,

$$\tilde{w}_t(i) = \sum_{j=0}^{R} w_t^g(i) s_t(i - j) \tag{11}$$

where $s_t$ is the shift weighting. To prevent the shifted weight from blurring, sharpening is applied

$$w_t(i) = \frac{\tilde{w}_t(i)^\gamma}{\sum\limits_j \tilde{w}_t(j)^\gamma} \tag{12}$$

Then, the memory is updated as follows,

$$M_t^{erased}(i) = M_{t-1}(i)\left[1 - w_t(i) e_t\right] \tag{13}$$
$$M_t(i) = M_t^{erased}(i) + w_t(i) v_t \tag{14}$$

where $e_t \in \mathbb{R}^D$ and $v_t \in \mathbb{R}^D$ are erase vector and update vector, respectively. The read value is computed using the same address weight as follows,

$$r = \sum_{i=1}^{N} w_t(i) M_t(i) \tag{15}$$

## G   Others

If we deliberately set the key dimension equal to the number of programs, we can even place an orthogonal basis constraint on the key space of NSM by minimizing the following loss,

$$l_{p_2} = \left\|\mathbf{M}_p.K\mathbf{M}_p.K^T - \mathbf{I}\right\| \tag{16}$$

where $\mathbf{M}_p.K$ and $\mathbf{I}$ denote the key part in NSM and the identity matrix, respectively.

Direct attention is one special case of key-value attention when the memory keys form orthogonal basis. When this happens, the generated key $k_t^p$ plays a direct role as the attention weight $w_t^p$. Thus, using key-value attention is more generic.

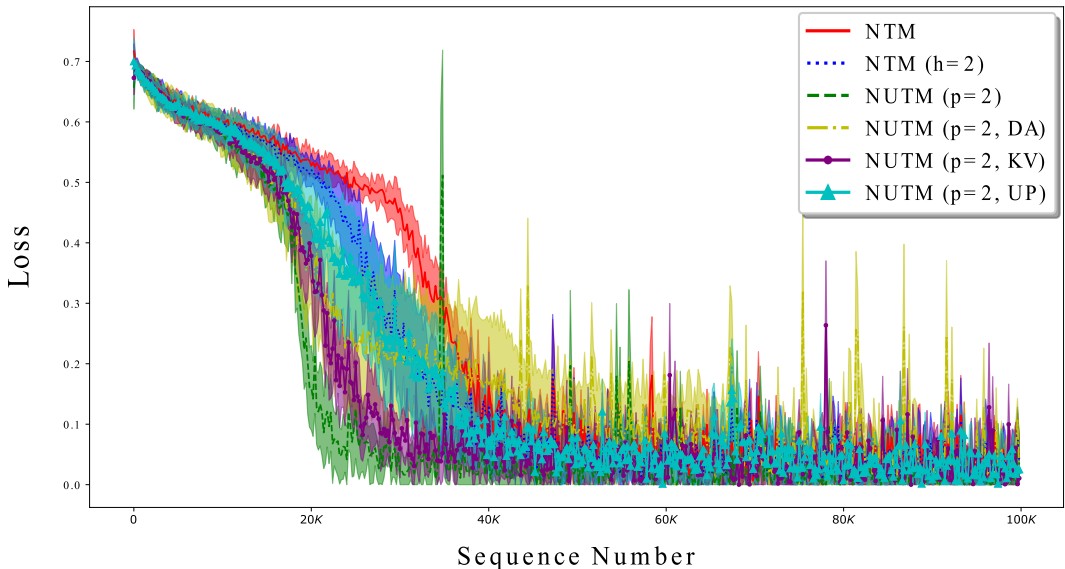

Figure 23: Learning curves on Associative Recall (AR) ablation study.

For all tasks, $\eta_t$ is fixed to 0.1, reducing with decay rate of 0.9.

Ablation study's learning losses with mean and error bar are plotted in Fig. 23.

