# OpenReview forum: "Neural Stored-program Memory"
_ICLR.cc/2020/Conference — Accept (Poster)_

### Official Review · AnonReviewer2 · 2019-10-18
**Official Blind Review #2**

**Rating:** 3

**Review:**

= Summary
A variation of Neural Turing Machines (and derived models) storing the configuration of the controller in a separate memory, which is then "softly" read during evaluation of the NTM. Experiments show moderate improvements on some simple multi-task problems.

= Strong/Weak Points
+ The idea of generalising NTMs to "universal" TMs is interesting in itself ...
- ... however, the presented solution seems to be only half-way there, as the memory used for the "program" is still separate from the memory the NUTM operates on. Hence, modifying the program itself is not possible, which UTMs can do (even though it's never useful in practice...)
- The core novelty relative to standard NTMs is that in principle, several separate programs can be stored, and that at each timestep, the "correct" one can be read. However this read mechanism is weak, and requires extra tuning with a specialized loss (Eq. (6))
~ It remains unclear where this is leading - clearly NTMs and NUTMs (or their DNC siblings) are currently not useful for interesting tasks, and it remains unclear what is missing to get there. The current paper does not try show the way there.
- The writing is oddly inconsistent, and important technical details (such as the memory read/write mechanism) are not documented. I would prefer the paper to be self-contained, to make it easier to understand the differences and commonalities between NTM memory reads and the proposed NSM mechanism.

= Recommendation
Overall, I don't see clear, actionable insights in this submission, and thus believe that it will not provide great value to the ICLR audience; hence I would recommend rejecting the paper to allow the authors to clarify their writing and provide more experimental evidence of the usefulness of their contribution.

= Minor Comments
+ Page 6: "As NUTM requires fewer training samples to converge, it generalizes better to unseen sequences that are longer than training sequences." - I don't understand the connecting between the first and second part of the sentence. This seems pure speculation, not a fact.

**Experience Assessment:**

I have published one or two papers in this area.

**Review Assessment: Checking Correctness Of Derivations And Theory:**

I assessed the sensibility of the derivations and theory.

**Review Assessment: Checking Correctness Of Experiments:**

I assessed the sensibility of the experiments.

**Review Assessment: Thoroughness In Paper Reading:**

I read the paper at least twice and used my best judgement in assessing the paper.

---

> ### Author Response · Authors · 2019-11-11
> **Response to reviewer 2**
>
> We thank the reviewer for your helpful comments. We address your concerns one by one as follows,
>
> “Hence, modifying the program itself is not possible, ...”. We want to clarify that our working program (the one that is loaded into the interface network) is modified across timestep and our basis program (the one that is stored in the program memory) is modified by backpropagation.
>
> “However this read mechanism is weak, ...”. Our program reading mechanism may be simple. Yet, it helps boost the performances of various MANNs. More importantly, it helps the models behave as designed (see Fig. 3 (a,b,c) and other figures in Appendix).
>
> “It remains unclear where this is leading ...”. Our method leads to a new class of MANNs that can store and query both the weights and data of their own controllers. One reason why MANNs maybe not useful is the lack of program memory. Without program memory, MANNs are inflexible, prone to perseveration and fail to simulate UTM. We aim to fix this weakness in this paper. Our attempts help MANNs improve their performance in various experiments including algorithmic, compositional, continual, few-shot learning and question-answering tasks (Sec. 4).
>
> “The writing is oddly inconsistent, ...”. As our method is generic and can apply to various MANNs, we cannot describe all memory access mechanisms of the chosen MANNs in the main manuscript. NSM's memory access uses key-value attention, which can be thought of as an extension of content-based attention presented in NTM or DNC. The main difference is that the value in NSM is the weight of a neural network, not the data as in other MANNs.
>
> “Overall, I don't see clear, actionable insights, ...”. In this paper, we provide the reader with a new insight into the simulation capacity of MANNs. We point out the missing part of current MANN literature and propose a more generic architecture that can simulate UTM. We have conducted 6 experiments to validate the benefit of our proposed architecture.
>
> Thank you for your minor comment. The sentence is based on our intuition and validated by Table 1. To make it less confusing, in the updated manuscript, we just report facts and do not assume any causality: “NUTM requires fewer training samples to converge and it generalizes better to unseen sequences that are longer than training sequences”.

---

> > ### Comment · AnonReviewer2 · 2019-11-14
> > **Discussion of Review#2**
> >
> > > "However this read mechanism is weak, ...". Our program reading mechanism
> > > may be simple. Yet, it helps boost the performances of various MANNs.
> >
> > You've cut the second half of that sentence, which was the relevant bit: "... and requires extra tuning with a specialized loss (Eq. (6))" - the point that I'm unhappy about here is that this loss term is quite unprincipled. The collapse of supposedly-separate memories into one "smeared" sum is a standard problem in continuous relaxations of memory reads; here you are circumventing that by penalising the cosine similarity between keys. But that's bad strategy when the size of the memory is growing, and it is not clear why keys should be far away in the first place.
> >
> > > “The writing is oddly inconsistent, ...”. As our method is generic and can apply
> > > to various MANNs, we cannot describe all memory access mechanisms of the
> > > chosen MANNs in the main manuscript.
> >
> > You _could_ describe one and make your paper more self-contained.
> >
> > Overall, I remain unconvinced that this submission is useful to people that are even slightly outside the immediate research field, and hence do not think that publication of it at a large conference is appropriate and thus will keep my recommendation at a weak reject.

---

> > > ### Author Response · Authors · 2019-11-15
> > > **Response to Discussion of Review#2**
> > >
> > > Thank you for your reply. Regarding your concern on the loss term (Eq. 6), we explain the necessity of this loss in the second paragraph of Sec. 3.1. We want to avoid the situation where changes in the query key (representing the context) do not lead to changes in the program. If the program keys are close to each others, the distances from the query key to the program keys are almost equal regardless of the value of the query key. This leads to uniform program attentions–one form of program collapse. Hence, program keys should be far away. You can refer to our response to the reviewer 3 (the second paragraph) to understand more on why separated program keys are usually good enough for our problem.
> > >
> > > It is clear from Eq. 6 that our extra loss is designed to separate the program keys. Our ablation study in Sec. 4.2 demonstrates that it helps improve the performance of NUTM. It should be noted that even without the extra loss, NUTM shows better performace than many other baselines (Fig. 2b).  Hence, our model does not depend heavily on tuning with a specialized loss. We introduce the extra loss as a nice-to-have feature for our model.
> > >
> > > We agree with you that when the number of the programs increases, computing Eq. 6 may be harder. However, in our setting, the program memory size is moderate (<6) and the key dimension is small (equal the number of programs). Hence, computing Eq.6 remains efficient. For applications that require many programs, we can reduce the computation cost by sampling pairs of program keys to impose the regularization. The experiments with large scale program memory is out of scope of this paper and will be left for future works.
> > >
> > > Regarding your comment on choosing one MANN to describe, we have considered it carefully. In Sec. 2, we describe the general operation of MANNs and providing a specific example may be a good idea. Due to page limit, we cannot put it in the main manuscript. However, we will add the description of a specific MANN (NTM) in the appendix of the final revision. Thank you for your suggestion.

---

### Official Review · AnonReviewer3 · 2019-10-20
**Official Blind Review #3**

**Rating:** 8

**Review:**

The authors discuss an interesting idea for memory augmented neural networks: storing a subset of the weights of the network in a key-value memory. The actual weights used are chosen dynamically. This is similar to having programs/subprograms in a classical computer. To the best of our knowledge, this idea was not previously studied, and it was a missing part of the study of memory augmented neural networks until now.

In the abstract authors mention a von Neumann architecture. However, the von Neumann architecture assumes that the program and the data are stored in the same memory, which is not the case for what the authors propose (a significant limitation is that the vector size of the memory doesn’t match the number of network weights). It is more reminiscent of the Harvard architecture. Please correct!

At the end of section 3.1 and in Eq 6 and 7 the authors claim that they use the regularization loss that makes the keys different to prevent programs from collapsing. However, this is only one way of collapsing. The other way is more reminiscent of what tends to happen in routing networks, where the controller network (here P_I) learns to attend to a single module (here memory slot) or attend to all of the modules with roughly equal weight. Are such collapses experienced during this work? If not, what could be the reason?

The main difference between routing networks and NSM is that in the former the selection is done in activation space, while in NSM it happens in weight space. Thus one should expect the same exploration/exploitation, transfer/inference tradeoffs, module collapse, etc to happen [1]. What makes it better/worse compared to these methods?

The method is mainly tested on methods used to test MANNs. However, because of its relation to the routing networks, it would also be interesting to compare them to [2] or [3] to see whether they suffer from the same problems.

Because the program distribution is rarely close to one-hot (all programs are contributing to some degree), could you please provide as a baseline a version where the program distribution is fixed and uniform?

In section 3.2, the last 2 sentences of the first paragraph, in one sentence the authors say “we store both into NSM to completely encode a MANN” and in the next they say “in this paper, we only use NSM to store W^C”. Please make this consistent.

In the last sentence of the first paragraph of section 3.2, the authors say “is equivalent to the Universal Turing Machine that can simulate any one-state Turing Machine”. Could you further clarify this?

In the second paragraph of section 3.2, the authors say “P_I simulates \delta_u of the UTM”. Shouldn’t \delta_u also include the state transition function, which is not included in P_I?

Several sentences mention “direct attention,” which is described as a neural network producing the weights directly. Why call this attention at all? Isn’t this a form of fast weights? Probably they should be called fast weights or dynamic links.

How does this relate to the original work on this? Note that the first end-to-end-differentiable systems that learn by gradient descent to quickly manipulate the fast weights of another net (or themselves) were published between 1991 and 1993 - see the references [FAST0-3a] [FAST5] [FASTMETA1-3] in section 8 of http://people.idsia.ch/~juergen/deep-learning-miraculous-year-1990-1991.html

The authors also claim to “When the memory keys are slowly updated, the meta-network will shift its query key generation to match the new memory keys and possibly escape from the local-minima”. Why? What drives the network to use new memory keys in this case?

In section 3.3, “P_I is a meta learner”. This is not the correct term here, because P_I is not leveraging old knowledge to learn faster (nor in improving the learning algorithm itself, nor in the recently more popular transfer learning sense), but it learns to select which weights to use.

In figure 2b, could you show the mean with std?

“states of the generating automation” should be automaton instead.

“As NUTM requires fewer training samples to converge, it generalizes better to unseen sequences that are longer than training sequences” - Why this indication of a causal relationship here? Why would requiring fewer training samples make it generalize better?

In the sequencing tasks (section 4.3), how does the network know which task to perform? Is the task label one-hot encoded and concatenated with the input? From Fig 13 and 14, it seems to be a fixed concatenation of inputs, without any indication which task is which, and then a fixed concatenation of the outputs is requested from the network. Could you clarify, please?

We found Figure 5 very confusing. At first it seemed like the X-axis is “time” (where we assumed the network is trained on a sequence of tasks C, RC, AR, PS), and the title above the subplots indicates for which task the performance is plotted over “time” (which in this case would correspond to the time steps after completion of the training phase indicated on the X-axis). However, the second subplot shows perfect RC performance after having been trained only on C. We probably misunderstood the plots: the title plot is the “time”, and the X-axis shows just the datasets (so the order is not important) - so the plots don’t show the performance on the specific dataset over the course of training as assumed initially. But if so, why are they connected by lines and why not use a bar plot? It would be more interesting to see “time” on the X-axis, so one can see how performance degrades while training on new tasks.

In Figure 5, what is the average performance supposed to show? If you average the performance over each training phase, the dataset trained first will yield better performance than the ones trained later. This is because the last trained one will have a performance near chance for most of the time it is measured, while the first one will have the best performance on the first measurement and will degrade somewhat because of catastrophic forgetting - but hopefully it will still be better than chance.

In Figure 5, numbers on the Y-axis: the first 10 and 100 place digits are misaligned.

In the last paragraph of section 4.4, “- over 4 tasks”, the - is confusing because it is not part of the formula. Please rewrite the formula in a different style or remove the -.

In section 4.5, the authors write that the best hyperparameters they found are p=2 for 5 classes and p=3 for 10. What could be the reason for p and the number of classes being so unrelated?

In section 4.6, “DNC is more powerful and thus suitable for NSM integration” - this suggests that the reason why it is suitable is that it is more powerful, but that is not the case since it was integrated into NTM, too.

In section 5, “they impede modularity”. Why? Maybe they don’t increase it, but why impede? “which looses modularity” - Why? The original RNN is not modular either. Why is this even less modular?

In figure 7 etc. it would be nice to have a title for the topmost subplot, too.

What is a dynamic n-grams task? Could you clarify or include a reference?

Could you clarify how the white-black-yellow-orange input charts should be understood (Fig 3d, Fig 15, etc)?

How to understand the preservation plots (Fig 3d)?

Additional references besides [FAST0-3a] [FAST5] [FASTMETA1-3] mentioned above:

[1] Rosenbaum et al, Routing Networks and the Challenges of Modular and Compositional Computation
[2] Chang et al, Automatically Composing Representation Transformations as a Means for Generalization
[3] Kirsch et al, Modular Networks: Learning to Decompose Neural Computation


We think this paper is quite interesting, and might improve our rating provided the comments above were addressed in a satisfactory way in the rebuttal.

Edit after rebuttal: score increased to 8. However, the authors should not forget to cite the original fast weight paper by Malsburg (1981). And one more thing: in the revised version they write "More recent works implement fast-weight using outer-products," citing papers from 2016-17, but this was actually first done in the 1993 paper they already cite.

**Experience Assessment:**

I have published in this field for several years.

**Review Assessment: Checking Correctness Of Derivations And Theory:**

N/A

**Review Assessment: Checking Correctness Of Experiments:**

I assessed the sensibility of the experiments.

**Review Assessment: Thoroughness In Paper Reading:**

I read the paper thoroughly.

---

> ### Author Response · Authors · 2019-11-11
> **Response to reviewer 3 (1/2)**
>
> We thank the reviewers for your detailed review. We address your concerns one by one as follows,
>
> Our proposal and Von Neumann Architecture share the stored-program concept. We agree that we should not have claimed that our model resembles Von Neumann Architecture in the abstract. We remove this phrase in the updated manuscript.
>
> We agree with you that program collapse can happen even when the program keys are different. However, under our MANN setting, given the program keys are different, the collapse or equal weight problems are not serious. As stated in Sec. 3.3, accessing the program memory is basically a regression problem that maps $c_{t}$ to $\xi_{t}=\left\{ k_{t}^{p},\beta_{t}^{p}\right\}$ . Assume that the program collapse or equal weight problems happen, NUTM now roughly becomes normal NTM with a single program. Even with a single program, NTM can learn to partition its state space to some degree (see Fig. 6 (a,c)). That provides enough training signals for the regressor (P_I in this case) to discriminate its input space $c_{t}$, which results in different $k_{t}^{p}$ when there is a significant change in the hidden state space. A small difference amongst $k_{t}^{p}$ will alleviate the program collapse or equal weight problems, which in turn, improves the clusters in the space of $c_{t}$ and then again alleviates further the program collapse or equal weight problems and so on.
>
> Thank you for pointing out interesting papers on modular learning. We discuss the relation between our model and module networks in the second paragraph of Sec. 5. Our approach is different since we do not use “hard” modules. Our model is different from routing networks since it not only selects but also interpolates the modules and the learning is fully differentiable. Moreover, in this paper, we aim to provide the first working simulation of UTM using MANN rather than focus on modular learning. Within the scope of our paper and due to page limit, we cannot go into details of analyzing problems related to modular learning or comparing our approach with other modular methods.
>
> Thank you for your suggestion on another baseline. A baseline with fixed and uniform program distribution will not adapt well with context changes. To verify that, we have added this baseline in the ablation study. The result can be found in Sec. 4.2.
>
> Thank you for showing us the inconsistency in the last sentence of the first paragraph of section 3.2. We have changed it to “we should store both into NSM ...”.
>
> One-state Turing machines do not use state transition function. They just read from and write to the memory tape. In other words, the interface network simulates a one-state Turing machine. Our proposed model stores only the interface network in the program memory. It can be interpreted as a Universal Turing machine that stores only one-state Turing machines. Hence, the UTM can simulate any stored TM by using the TM program.
>
> Your comment on the second paragraph of Sec. 3.2 is correct. P_I alone does not simulate \delta_u of the UTM. We have changed it to “Together with the RNN, P_I simulates \delta_u of the UTM ...”
>
> In the third paragraph of Sec. 3.2, we define “direct attention” as generating the weight $w_{t}^{p}$ directly without matching $k_{t}^{p}$ with $\mathbf{M}_{p}\left(i\right).k. w_{t}^{p}$ is then used to weight (attend to) the program memory (Eq. 5). The generated weight $w_{t}^{p}$ is not the weight of a neural network. Hence, we do not call it fast weight.
>
> In the third paragraph of Sec. 5, we describe the relationship between our programs and fast-weights. The main difference is that our “fast-weight” is composed by interpolations of a set of programs. The program is slowly updated by back-propagation. Moreover, the “fast-weight” in our paper is motivated by UTM and we focus more on simulating UTM. Amongst additional references on fast-weight you recommend, we realize that [FAST3] and [FASTMETA2] are most related. We have revised the third paragraph of Sec. 5 to include these references.
>
> Regarding our sentence “When the memory keys are slowly updated, ...”, we do not mean the network needs to use new memory keys. The network just needs to change the way it generates the query key to match the change of program memory keys. This is inevitable since the gradient backpropagated to P_I contains $\mathbf{M}_{p}\left(i\right).k$ terms.
>
> Regarding our sentence “P_I is a meta learner”, a program can be thought of as the knowledge about a learnt task/subtask. In an ideal setting, P_I stores the knowledge in separated slots of the program memory. Hence, the knowledge is preserved and can be utilized by P_I when it handles new task/subtask. It is fair to say P_I is a meta learner.
>
> In Fig. 2(b), due to small size, adding std make the plot hard to see. We add a separate bigger plot of Fig. 2(b) with mean and std in Appendix Fig. 23 in this revision.

---

> > ### Author Response · Authors · 2019-11-11
> > **Response to reviewer 3 (2/2)**
> >
> > We have changed “automation” to “automaton” in this revision. Thank you for correcting our mistake.
> >
> > Regarding our sentence “As NUTM requires fewer training samples to converge, ...”, we admit that cannot prove the causality. This is based on intuition and empirical results (Table 1). To make it less confusing, in the updated manuscript, we just report facts and do not assume any causality: “NUTM requires fewer training samples to converge and it generalizes better to unseen sequences that are longer than training sequences”.
> >
> > In the first paragraph of Sec. 4.3, we explain “the order of subtasks in the sequence is dictated by an indicator vector put at the beginning of the sequence”. We do not use task label. Instead, we use order label to tell the model the order of subtasks. For example, in C+RC tasks, we use an indicator [0] to indicate the order C–RC and [1] for RC–C. The indicator vector is then padded to the same size with other input vectors and put at the beginning of the sequence. The order is randomized so the concatenation of inputs is not fixed.
> >
> > In Fig. 5, we follow the standard report format in [1] so we do not use bar plot. X-axis is across tasks, not time. We explain in the caption: “bit accuracy on four tasks after finishing a task” and the title of each subplot indicates the finished task. Adding training steps will further complicate the plot so we decide to keep it as is.
> >
> > Your understanding of the average subplot in Fig. 5 is correct. The average performance tends to lower for later tasks. However, because the tasks are correlated (e.g., most tasks often share the same “write all” strategy in the encoding phase), the performance on later tasks can be improved even before we train them with their data (a form of transfer learning). For example, in the first subplot of Fig. 5, after training C, the performances on RC are improved moderately. Thus, we think it is useful to compute the average performance to measure both catastrophic forgetting and knowledge transferring effects.
> >
> > In this revision, we have fixed the misalignment of the y-axis in Fig. 5 and remove - in “- over 4 tasks”. Thank you for pointing out these errors.
> >
> > In Sec. 4.5, increasing the number of classes normally requires more adaption. For example, each class of images may need a specific program to store into the data memory thus 10 classes should have more programs than 5 classes. However, as the classes are always correlated (e.g., sharing visual features), the number of optimal programs may be actually less than the number of classes. Also, learning with many programs may be uneasy. Hence, we cannot give you an exact reason. Only by tuning the number of programs, we know the ones that work best.
> >
> > In Sec 4.6, our complete sentence is “DNC is more powerful and thus suitable for NSM integration to solve non-algorithmic problems such as question answering”. We only assume DNC is more powerful than NTM in question answering and suitability here applies to only this task. We intend to show that our NSM can work with various MANNs. That is why we integrate NSM into both NTM and DNC.
> >
> > In Sec. 5, Multiplicative RNN looses modularity because it no longer keeps the form of multiple slow-weights as its precursor-Tensor RNN does. We agree with you that using “impede” is a bit extreme. we have changed to “do not support” in this revision.
> >
> > Dynamic N-Grams is introduced with other NTM's synthetic tasks in Sec. 4.1. It is from [2].
> >
> > In Fig. 3, black is bit 0, white is bit 1 in vector data. Orange is prediction error (noted in the caption). In Fig. 15 and some other images, because data vectors not only include value 0-1, but also other float values (e.g., priority score), the color scale is automatically changed. Basically, in priority sort task, yellow is prediction error, and orange is bit 1. We have noted this inconsistency in this revision.
> >
> > Perseveration in Fig. 3d is explained in the second paragraph of Sec. 4.3. We add more details here. As you can see, there are two input patterns: the first is for Repeat Copy and the second Copy. Our example shows that NTM only executes Repeat Copy with the first pattern. It is easier to look at the reading behaviors. NTM repeatedly reads from the memory slots (0-15) containing the first pattern (its reading is not sharp, which leads to incomplete reconstruction of the first pattern). After 11 repeats, NTM should have changed to memory slots (16-30) to execute Copy on the second pattern. However, it fails to do so even when its writing during encoding is relatively good (both patterns are stored correctly, see Fig. 17).
> >
> > Reference:
> >
> > [1] Friedemann Zenke, Ben Poole, and Surya Ganguli. Continual learning through synaptic intelligence. In Proceedings of the 34th International Conference on Machine LearningVolume 70, pp. 3987{3995. JMLR. org, 2017
> >
> > [2] Alex Graves, Greg Wayne, and Ivo Danihelka. Neural turing machines. arXiv preprint arXiv:1410.5401, 2014

---

### Official Review · AnonReviewer1 · 2019-10-22
**Official Blind Review #1**

**Rating:** 6

**Review:**

## Update

I am changing my score to 7 (not adjusting the discretised score in the openreview interface though)





Summary: The authors present the Neural Stored-program Memory (NSM), an architectural addition which is suitable for a large class of existing MANN models (NTM, DNC, LRUA etc). The model contains a second memory of key value pairs, which on each timestep can be read via cosine distance similarity on the keys, and for which the values contain all the weights necessary for the Interface Network (IN). The IN also receives the recurrent controller output, and produces an interface vector which is used to read or write to the regular memory. The definition is sufficiently general to allow the computation between an interface vector and the regular memory to be done according to NTM/DNC/etc schemes.

NSM allows a MANN to effectively switch programs at every timestep, more closely matching classical computers and the Universal Turing Machine. The authors include a wide range of experiments, both showing faster learning on standard NTM tasks, and also introduce "Sequencing tasks" for which a single episode contains multiple instances of the standard NTM tasks in sequence. In principle this should allow the whole system to learn different programs for the subproblems, and dynamically switch between them as soon as the task changes. In principle a standard NTM could learn to do many of these tasks in a row, but the authors show that even when an NTM combined with NSM (denoted NUTM) has fewer trainable weights in total than a plain NTM, for some task sequence combinations the NUTM learns much faster.

Experiments on continual learning, few shot learning and text based question answering back up the wide applicability of this technique, with many strong results including new SOTA on bAbI.


Decision: Accept. Key reasons are that this is a relatively straightforward application of hypernetworks within a recurrent controller, which both has appealing justifications in the context of both Von Neumann and Turing models of computation. This simplicity is a positive, and the authors make a convincing argument that NSM can be applied to any MANN solving a sufficiently complex problem. A secondary reason is the extensive evaluation & hyperparam details, and while I do have some minor points on which I think the paper could be clarified (see below) I think this is overall a very nice paper. With the clarifications below addressed, I would give this paper a 7 but from the options I have available to choose from, 6 is the best fit for the current manuscript.

Supporting arguments: The visualization of 'program distribution' shows extremely clear phase changes as the underlying task changes - both within a single task (the reading phase vs writing phase for repeat copy) and across task boundaries (copy -> associative recall). Combined with the learning curves / results in the various tables, it is to me clear that the model is performing as designed.

Despite the name of the "Meta Network", and having a vague flavour of metalearning, the model does not require any elaborate "backprop through the inner training loop" of MAML et al, which is a benefit in my opinion.




Additional feedback: Some of the experiments could be slightly more convincing - particularly Figure 5 which is lacking error bars. In my experience these architectures can have relatively high variance in performance, compared to other supervised domains, as evidenced by the spicy learning curves even in Figure 2 a). Error bars across multiple runs for figure 5 would be good, particularly for the points where the lines are close (eg after training on PS, the performance for C and RC is close for both models).

The formatting of some figures and graphs could be improved:
* Figure 1 - the graph could use some more text labels, rather than mathematical notation which needs to be referred to below. The colours are also slightly confusing - the program memory has slightly different shades of orange for the keys, slightly different shades of green for the values, whereas the regular memory has a slightly wider variety of colours. I was not sure at first whether this should be interpreted as indicating something important. Additionally, the value read from the NSM is reshaped to a 2x4 shape with various shades of blue, which then becomes the weights for a pink network? I think the colour adds little and may confuse people. There are some other issues, such as the $r_t$ value which should really come from the main memory as a result of the interface vector. With the supporting text, understanding the system is not hard, but I feel another pass over the diagram would benefit the camera ready version - consider ditching colour entirely, unless there is going to be some consistent meaning to things being the same colour vs not. Text labels with the various arrows (eg "vector used to lookup in NSM", "used as network weights for Interface Network") may improve clarity. $c_t$ should also be labelled on the diagram.

* The y axis scale in figure 5 is very confusing - it took me several looks before I noticed that it goes from 50% to 100%, due to each number having digits in different vertical positions and different font sizes

* Figure 2 y axis scale is a bit too small to easily read.

* Figure 3: both read and write locations are shown on a single plot, but the green line that separates them is lamost unreadable on a printout. The task dependent, presumably manually chosen, approach to picking where this visualisation toggle should be made is a bit arbitrary - I would prefer to see read and write locations as separate subplots, as in the appendix.


I found the exact details of number of programs versus number of heads (and the type of those heads) a bit confusing. In www.github.com/deepmind/dnc the code has 'number of read heads' and 'number of write heads' being two independently set integer parameters. This paper refers to "$R$ control heads", but I am not exactly clear on how these are divded between read and write duties. Algorithm 1 references that write heads return zero on line 8 but not other mention of the two mutually exclusive types is made. The text towards the end of section 4.1 refers to the "no-read" strategy being assigned mostly to one program - this makes it sound like each program can (softly?) interpolate between reading or writing (or both?). The start of appendix B shows program distribution for read and write heads separately, but this then begs the question of what is happening in the examples for which only one program distribution is shown (eg Figure 3) - clearly we need to read and write for all of these tasks, so is one head with two programs doing both simultaneously? In the interests of reproducability, clarification here is essential.

Separately, the decision of having a different program memory per control head is interesting - it's not obvious to me why this would be necessary, surely one program memory would be sufficient as long as thre is a different $P_{I,n}$ (alg 1 line 5) network to choose a different program for the head? It would be good to see a line added to the paper justifying this choice.

It seems like not all training hyperparams are specified in the appendix - eg the settings specified in Table 9 only apply to the few shot learning task, additional hyperparams are specified for bAbI, but I cannot see the training hyperparams for the experiments in sections 4.1 - 4.4.

Minor correction:

"As Turing Machine is finite state automata" -> "As Turing Machines are finite state automata"


**Experience Assessment:**

I have published one or two papers in this area.

**Review Assessment: Checking Correctness Of Derivations And Theory:**

N/A

**Review Assessment: Checking Correctness Of Experiments:**

I assessed the sensibility of the experiments.

**Review Assessment: Thoroughness In Paper Reading:**

I read the paper at least twice and used my best judgement in assessing the paper.

---

> ### Author Response · Authors · 2019-11-10
> **Response to reviewer 1**
>
> We thank the reviewer for your thoughtful comments. We address your concerns one by one as follows,
>
> We agree that error bars are necessary for Fig. 5. We have re-run the experiments 5 times to report the results with error bars in this revision. We also fix the misalignment of the y-axis in Fig. 5 and scale-up y-axis in Fig. 2a. Thank you for pointing out these errors.
>
> In this revision, we have made Fig. 1 clearer by synchronizing the colors and adding a legend box to explain the components of our model. We also agree that the read value should come from the memory M. Thank you for your suggestion.
>
> In Fig. 3, we plot reading behaviors for the repeat copy (a) and C+AR (c), and writing behavior for the priority sort (b). The green line separates the encoding and decoding phase. Due to space limit, we cannot plot both reading and writing behaviors for each task as in the appendix.
>
> We want to make it clear that $R=R_{r}+R_{w}$ where $R_{r}$ and $R_{w}$ are the number of read and write heads, respectively. Each head, either read or write, is associated with a memory() function. The implementation of memory() depends on the type of heads (read or write) and MANNs. The duties of read or write are specified inside the function memory(). In this revision, we make it clearer in Line 8 Algo. 1. Moreover, we modify all hyperparam tables in the Appendix to clarify the number of read and write heads used in our experiments. The “no-read” strategy text describes Fig. 3 (a), which is only about reading behavior in repeat copy. The “orange” program only represents the strategy for the read head in this task. There is no writing program in this description. In this task, there are 1 read head and 1 write head, each of which has 2 programs. In total, we have 4 programs and we only plot two of them (orange+blue) in Fig. 3 (a). We are sorry that we cannot include visualization for writing program distribution due to space limit. In general, a program cannot do both reading and writing because the set of parameters for reading and writing (the interface vectors) are different. In Fig. 8, the orange program in the last subplot (for write head) is different from the one in the third subplot (for read head). To make it clear, we add head types in Fig. 3 in this revision.
>
> Your question on why not one program memory is interesting. Assume that you have R heads and want to apply P programs per head. If you use single program memory, there should be $R\times P$ program slots and you have to attend to all of these slots. That is, all slots interact together. In our approach, we make use of an inductive bias that the subset of programs for each head should be separated from each other. One NSM per head is to ensure the programs for one head do not interfere with other heads and thus, encourage functionality separation amongst heads (otherwise, we may not need multiple heads). Also, as each program memory now only has P program slots, program attention should be easier. It is similar to hierarchical attention. You attend to the program memory first, then attend to the program slot in that program memory. In our case, the first attention is straightforward due to the inductive bias (we associate the head with the program memory directly). We have added a brief explanation in the updated manuscript (the fourth paragraph of Sec. 3.2).
>
> Hyperparams for Sec. 4.1, 4.2 are in Table 4, Sec, 4.3 in Table 6. We only missed hyperparams for Sec. 4.4. Thank you for pointing this out. In this revision, we list hyperparam table for Sec. 4.4 in Table 8.
>
> Your minor correction is correct. Thank you for correcting us.

---

> > ### Comment · AnonReviewer1 · 2019-11-15
> > **Response to Rebuttal**
> >
> > Thanks for your rebuttal and updated draft. The updated hyperparameters look complete to me, and I appreciate the more explicit split between read and write heads. I note that in what I presume is the most recent version (there are 3 unlabelled updated drafts in the system) there isn't a whole lot of changes to the text - I would maybe advise to put a bit more of the explanations you've given me back into the paper. For example, making more explicit in the text (the new caption is good) that figure 3 a) only shows the behaviour of the read head - maybe "Examining individual heads, we observe two program usage patterns..."
> >
> > I appreciate the space constraints make it hard to show everything, and the green line to separate read and write phases is necessary - but I would highly advise you to make it 10x wider, as it's hard to read on both paper and some screens.
> >
> > Given the changes I am raising my score to a 7

---

### Author Response · Authors · 2019-11-11
**Summary of revision**

We would like to thank all the reviewers for their constructive feedback. In this revision, major changes include (1) update Fig. 1 with simple colors and more text labels, (2) add uniform program baseline in the ablation study with new result in Fig. 2b, and add a bigger version of Fig. 2b with error bars in Appendix Fig. 23 (3) fix misalignment and add error bars for Fig. 5. Other specific adjustments are mentioned in the responses below.

---

### Decision · Program_Chairs · 2019-12-19

**Decision:**

Accept (Poster)

**Comment:**

This paper presents the neural stored-program memory, which is a key-value memory that is used to store weights for another neural network, analogous to having programs in computers. They provide an extensive set of experiments in various domains to show the benefit of the proposed method, including synthetic tasks and few-shot learning experiments.

This is an interesting paper proposing a new idea. We discuss this submission extensively and based on our discussion I recommend accepting this submission.

A few final comments from reviewers for the authors:
- Please try to make the paper a bit more self-contained so that it is more useful to a general audience. This can be done by either making more space in the main text (e.g., reducing the size of Figure 1, reducing space between sections, table captions and text, etc.) or adding more details in the Appendix. Importantly, your formatting is a bit off. Please use the correct style file, it will give you more space. All reviewers agree that the paper are missing some important details that would improve the paper.
- Please cite the original fast weight paper by Malsburg (1981).
- Regarding fast-weights using outer products, this was actually first done in the 1993 paper instead of the 2016 and 2017 papers.